# Repurposing host-guest chemistry to sequester virulence and eradicate biofilms in multidrug resistant *Pseudomonas aeruginosa* and *Acinetobacter baumannii*

Christopher Jonkergouw [1] ✉, Ngong Kodiah Beyeh [2,3], Ekaterina Osmekhina [1], Katarzyna Leskinen[4], S. Maryamdokht Taimoory[5,6], Dmitrii Fedorov[1], Eduardo Anaya-Plaza [1], Mauri A. Kostiainen [1], John F. Trant [5], Robin H. A. Ras [1,3], Päivi Saavalainen[4,7] ✉ & Markus B. Linder [1] ✉

The limited diversity in targets of available antibiotic therapies has put tremendous pressure on the treatment of bacterial pathogens, where numerous resistance mechanisms that counteract their function are becoming increasingly prevalent. Here, we utilize an unconventional anti-virulence screen of host-guest interacting macrocycles, and identify a water-soluble synthetic macrocycle, Pillar[5]arene, that is non-bactericidal/bacteriostatic and has a mechanism of action that involves binding to both homoserine lactones and lipopolysaccharides, key virulence factors in Gram-negative pathogens. Pillar[5]arene is active against Top Priority carbapenem- and third/fourth-generation cephalosporin-resistant *Pseudomonas aeruginosa* and *Acinetobacter baumannii*, suppressing toxins and biofilms and increasing the penetration and efficacy of standard-of-care antibiotics in combined administrations. The binding of homoserine lactones and lipopolysaccharides also sequesters their direct effects as toxins on eukaryotic membranes, neutralizing key tools that promote bacterial colonization and impede immune defenses, both in vitro and in vivo. Pillar[5]arene evades both existing antibiotic resistance mechanisms, as well as the build-up of rapid tolerance/resistance. The versatility of macrocyclic host-guest chemistry provides ample strategies for tailored targeting of virulence in a wide range of Gram-negative infectious diseases.

The scope of current antibiotic targets is very limited, with the vast majority of approved treatments targeting either DNA synthesis, protein synthesis, membrane integrity or cell-wall biosynthesis, all of which are counteracted by numerous and increasingly prevalent resistance mechanisms. The ensuing imminent global health crisis has been documented from several perspectives in recent reportings[1–3] and has prompted the World Health Organization to create a priority pathogen list. Gram-negative bacteria, which have an outer membrane

[1]Aalto University, School of Chemical Engineering, Department of Bioproducts and Biosystems, Kemistintie 1, 02150 Espoo, Finland. [2]Oakland University, Department of Chemistry, 146 Library Drive, Rochester, MI 48309-4479, USA. [3]Aalto University, School of Science, Department of Applied Physics, Puumiehenkuja 2, Espoo, Finland. [4]University of Helsinki, Translational Immunology Research Program, Haartmaninkatu 8, 0014 Helsinki, Finland. [5]University of Windsor, Department of Chemistry and Biochemistry, Windsor, ON N9B 3P4, Canada. [6]University of Michigan, Department of Chemistry, Ann Arbor, MI, USA. [7]Folkhälsan Institute of Genetics, Folkhälsan Research Center, Helsinki, Finland. ✉e-mail: chris@jonkergouw.nl; paivi.saavalainen@helsinki.fi; markus.linder@aalto.fi

(OM) that acts as a formidable barrier against a large number of standard-of-care antibiotics, represent the majority of pathogens on the priority list[4], with carbapenem-resistant *Acinetobacter baumannii* and *Pseudomonas aeruginosa* featuring at the top[5]. This pressing global challenge emphasises the need for the discovery of therapies with unique molecular scaffolds and/or mechanisms of action that evade existing resistance mechanisms and are less likely to lead to drug resistance[6].

A promising alternative to the limited antibiotic scope is the targeting of bacterial virulence factors, which play a crucial role in the establishment and progression of infections[6]. Potential targets have included pore-forming toxins[7], homoserine lactone signalling molecules[8–10], bacterial biofilms[11,12], metal chelation[13] and the bacterial OM[14]. Macrocyclic cavity-containing molecules are intensively explored in host-guest chemistry by supramolecular chemists, due to their tunable molecular preferences. Specifically, varying steric and conformational preferences enable the differential encapsulation of specific guests within their inner cavities[15]. The core structure can also be readily modified or enhanced with functional groups to add secondary functionalities[16], with several examples emerging of simultaneous interactions taking place[17,18]. Recent efforts have also explored the application of functionalized macrocycles towards antibacterial therapies, for both Gram-positive[19] and Gram-negative[20] pathogens, including biofilm targeting compounds[21]. However, the relationship between structure, function, and mechanism of action remains poorly understood.

Homoserine lactones (HSLs) are small diffusible signal molecules produced, released, and sensed by Gram-negative bacteria in a process called quorum sensing (QS) and are used to coordinate colony-wide responses including biofilm formation, production of exotoxins and surfactants, motility, and nutrient scavenging molecules[22,23]. As such, they form an attractive target for, and have previously been explored as, anti-virulence strategies[24,25]. Lipopolysaccharides (LPS) form a major component of the outer leaflet of the OM in gram-negative bacteria, posing as a formidable barrier against intracellular antibiotics[26].

In this work, we explore the interactions between the synthetic macrocycle Pillar[5]arene (P[5]a) and two distinct virulence factors: HSLs and LPS. We investigate the interaction between P[5]a and HSLs and how this shuts down virulence-controlled responses in bacterial pathogens. Similarly, we study the interactions between P[5]a and LPS and how this affects the function of LPS in the outer membrane, resulting in increased penetration of antibiotics across the OM barrier. Collectively, we investigate how P[5]a suppresses bacterial virulence in extensively drug resistant isolates and observe that in the absence of virulence factors, viability of lung epithelial cells recovers dramatically. The combined interactions of P[5]a with bacterial virulence factors provide a promising strategy against antibiotic-resistant infections, either alone or in combination with standard-of-care antibiotics.

## Results

### Antivirulence screen for macrocycle-HSL host-guest interactions

To identify possible host-guest interactions between cavity-containing macrocycles and bacterial signalling molecules, we set up an unconventional screening of a set of macrocycles (Supplementary Table 1) for affinity to six different HSLs (Fig. 1a, b). Although previous efforts already established a link between size of the internal cavity and binding affinity[9], little is known related to the affinities, preferences and determinants of host-guest interactions with HSLs. To gain a better understanding, and assuming that their cavity size plays an important role in determining HSL affinity, widely varying types of macrocycles were selected, primarily to represent a diverse range of inner cavity sizes (1.5 – 11.7 Å) and molecular weights (172 – 2.260 g/mol). Similarly, HSLs were selected to represent diversity in their acyl

chains, including a branched carbon, carboxyl groups and hydroxyl groups. We also screened for binding to pC HSL, containing a terminal *para*-phenol.

For the screening, we developed a fluorescent *E. coli* reporter system capable of detecting the six different HSLs. Each receptor dimerizes and binds to the promoter linked to EGFP upon addition of its cognate HSL (Fig. 1c, Supplementary Fig. 1, Supplementary Table 2). We screened for macrocyclic virulence inhibitors with a strong signal ratio for each of the HSLs ($I_1/I_0$ fluorescence) and minimal effects on bacterial viability and growth rate ($I_1/I_0$ optical density, ± 20%, grey dashed lines). We identified several macrocycle-HSL combinations of interest with a fluorescence signal ratio over 50% and minimal effects on growth rate, highlighted in blue circles (Fig. 1d, Supplementary Fig. 1a-e). All host-guest combinations with increased binding, fall within a limited range of inner cavity sizes, from 3 Å for 4-sulfocalix[4] arene (4-Sc[4]a) to 7.8 Å for β-Cyclodextrin (β-CD), suggesting that the inner cavity plays a role in the binding. Among the macrocycles tested, P[5]a showed the highest concentration-dependant signal ratio. Even though P[5]a shows binding with all six HSLs, it ranges from almost no binding to the C4 HSL (18%), with a very short acyl chain, to strong binding to the 3-oxo-C12 (64%) and 3:OH:C14 HSL (96%), both with prolonged acyl chains (Fig. 1e).

P[5]a had no observable effect on growth rate in *E. coli* and a viability assessment after 24 h growth shows that all concentrations tested (up to 2.5 mM), were well tolerated (Fig. 1f, Supplementary Fig. 2a-e). Looking more closely at the structure of P[5]a (Fig. 2a), shows that it is a modified pillar[5]arene with a lipophilic core and a hydrophilic periphery. It contains a cavity which is hydrophobic (4.6 Å), and a symmetrically substituted polycationic rim. Fluorescence microscopy and flow cytometry analysis show that in a combined microbial community, consisting of four *E. coli* sensor strains each with a unique fluorescent protein that detects a different HSL, P[5]a selectively binds the 3-OH-C14:1 HSL with an elongated acyl chain, resulting in a $10^2$ reduction in fluorescence of the *pCin*-EGFP sensor strain (Fig. 2b-d). In contrast, expression of the other three sensor strains remained largely unaffected by the addition of P[5]a, suggesting that the host-guest interactions could be utilized for specific targeting of bacteria with longer acyl chains. The selectivity, we propose, can be a beneficial approach, to target pathogens more specifically in comparison to broad-spectrum antibiotic activity. In various environments, such as the human microbiome, a diverse microbial gut community is essential for the health of its host. Antibiotic treatment can have a detrimental impact on the microbiome diversity, deteriorating the overall health of the host, while also leaving them far more vulnerable to *Clostridium difficile* infections.

### P[5]a inhibits QS-associated virulence in carbapenem-resistant *P. aeruginosa* and *A. baumannii*

To understand the possible implications of the host-guest interactions between the macrocycles and HSLs in a biological context, we continued the validation against the clinically relevant *P. aeruginosa* and *A. baumannii*, which both deploy long-chain HSLs (Fig. 2b). Extensive previous work on these pathogens has elucidated the function of HSLs in QS, an essential regulatory mechanism of collective virulence in these pathogens[27–29]. We screened the macrocycles that displayed the highest affinity, (i.e. P[5]a, α-cyclodextrin (α-CD), β-CD, and 4-Sc[4]a), for their effects on the production of pyocyanin in *P. aeruginosa* isolate PAO1. Pyocyanin is an exotoxin with an observable strong green color that is under direct regulation of QS, used as a surrogate measure of virulence (Fig. 2e). The addition of P[5]a resulted in a concentration-dependent reduction in pyocyanin levels, starting with concentrations of P[5]a as low as 10 μM, with complete suppression at 1 mM and above. In contrast, macrocycles α-CD, β-CD and 4-Sc[4]a had no effect on pyocyanin levels (Supplementary Fig. 3a-c). The incubation of macrocycles with purified pyocyanin confirms that P[5]a inhibits the

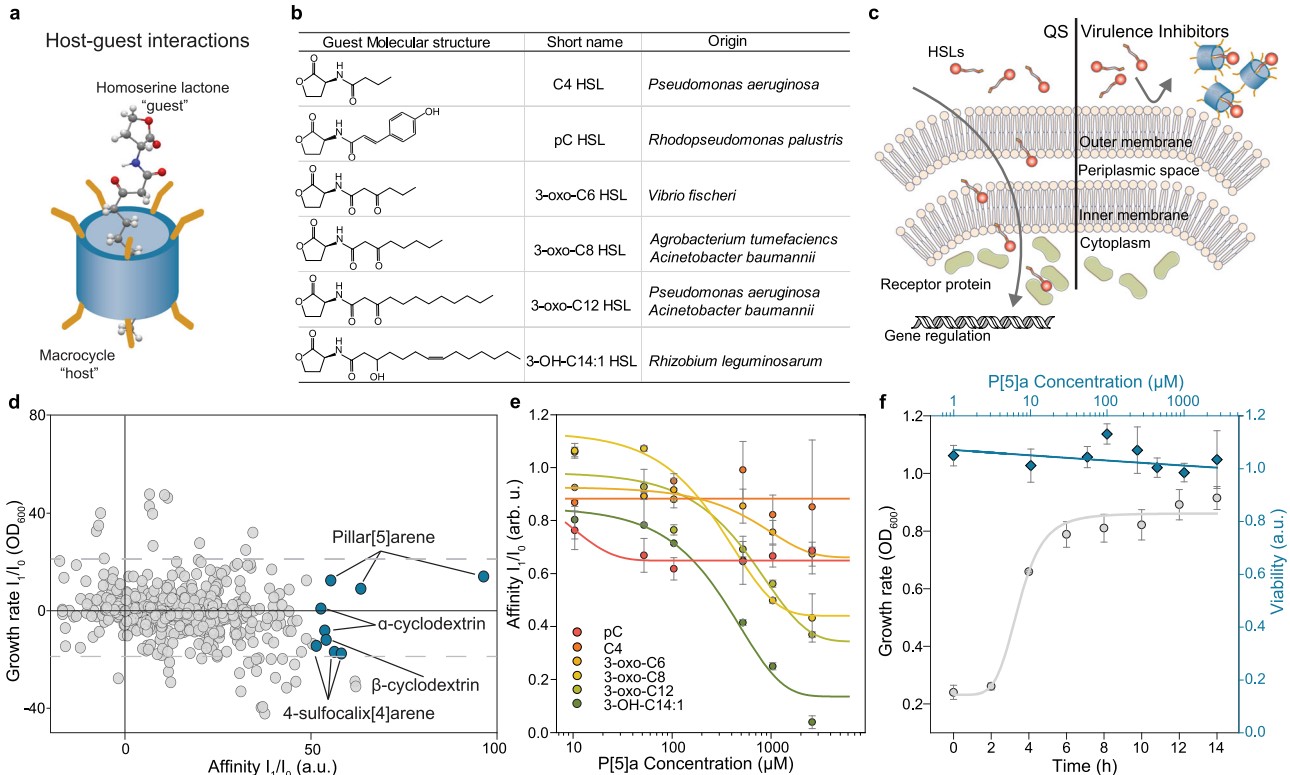

**Fig. 1 | Host-guest interactions between macrocycles and bacterial homoserine lactones selective for HSLs with prolonged acyl chains. a** Schematic of the interaction between macrocyclic "hosts" with HSL "guests". **b** The molecular structures, abbreviated names and bacterial species of all QS HSLs used in this study. **c** Schematic representation of HSL quorum sensing in Gram-negative bacteria (left), along with a virulence inhibition approach (right). In short, HSLs produced by select individuals diffuse freely through the outer and inner membranes and get sensed by intracellular receptor proteins in other community members, promoting the production of isogenic HSL and a synchronized regulation of activities. Applied virulence inhibition strategy, where the interaction between the HSLs and macrocyclic host prevents the recognition of HSLs by its cognate receptor. **d** Screening for the interaction between HSLs as seen in (**b**) and a set of macrocycles reveals several combinations where the macrocycle interacts with the HSL and where the growth rate in E. coli ($I_1/I_0$ $OD_{600}$) remains mostly unaffected

(threshold, grey dashed lines, ±20%). The interaction is detected through a HSL biosensor, where the cognate receptor is expressed, and the promotor binding region is fused to GFP. Interactions between macrocycles and HSLs are observed through a decrease in affinity ($I_1/I_0$). **e** Concentration-dependent effects of the strongest interacting macrocycle, P[5]a, on the fluorescent expression in E. coli for each of the six HSLs, indicates increased affinity towards HSLs with longer acyl chains. The data represent average values of biological replicates ± s.d ($n = 3$ per group). **f** Effect of P[5]a on the growth rate and viability of E. coli. E. coli incubated with highest tested concentration of P[5]a, 2.5 mM, shows a standard growth curve, including lag, log and stationary phase. A BacLight bacterial viability assessment shows cell viability in all tested concentrations of P[5]a after 24 h of exposure and no concentration-dependent cell death. The data represent average values of biological replicates ± s.d ($n = 3$ per group).

production of pyocyanin rather than interacting with existing toxin molecules (Supplementary Fig. 3d). The concentration-dependent effects of P[5]a on pyocyanin production in PAO1 strongly correlates with the effects of P[5]a on the quorum sensing biosensor in E. coli (Supplementary Fig. 3e).

P[5]a had comparable effects on biofilm formation in PAO1, also strongly associated with HSL-regulated QS, as indicated by the purple stain (crystal violet), with a similar concentration dependency as observed in the pyocyanin assessment (Fig. 2f). Because of the increasing pressure of resistance on antibiotic therapy, we tested the effectivity against top priority carbapenem- and third/fourth-generation cephalosporin-resistant *Pseudomonas aeruginosa* and *Acinetobacter baumannii* isolates (Fig. 2g, h, and Supplementary Data 1). In all cases, we observed a statistically significant decrease in biofilm formation. The effectivity of P[5]a against strains *P. aeruginosa* 2798 and *A. baumannii* 5542, 5707 and 5568 (Fig. 2g, h) is of particular importance, as these "superbug" pathogens are resistant to nearly all clinical treatments (Supplementary Data 1).

## P[5]a binds prolonged acyl chain HSLs inside its inner cavity
In order to confirm that the observed virulence inhibition-responses are a result of the interactions between P[5]a and the HSLs within the

core cavity, and to clarify the observed preferences for long-branched HSLs, we utilized a dye displacement assay (Fig. 3a)[30]. The binding of methylene orange (MO) to P[5]a results in a shift of the absorption maximum from 470 nm to 392 nm, as reported (Fig. 3b, left)[18,31] Subsequent addition of HSLs releases the MO by competitive binding (Fig. 3b, right). The resulting recovery of MO absorption at 470 nm allows to determine the binding affinity of HSL-P[5]a (Supplementary Fig. 4a-c).

The dissociation constant ($k_D$) between P[5]a and the HSLs was determined, allowing us to rank the binding affinity of HSLs into the P[5]a cavity (3OH-C14:1 ≥ 3-oxo-C12 > 3-oxo-C8 » 3-oxo-C6-pC-C4) (Fig. 3c, Table 1, Supplementary Fig. 5). This is in agreement with the hydrophobicity decrease expected from the decrease in acyl chain length, as observed in Fig. 1e. The addition of NaCl affected the association constants between MO and P[5]a, whereas the dissociation constants between P[5]a and the HSLs remained unaffected (Fig. 3c, Table 1, Supplementary Fig. 6c). This suggests that electrostatic interactions do not influence the binding of guests inside the core cavity.

To better understand which chemical interactions are involved and how structural features influence binding preferences, we used NMR (Fig. 3e). The $^1H$ NMR results confirm the stronger interaction

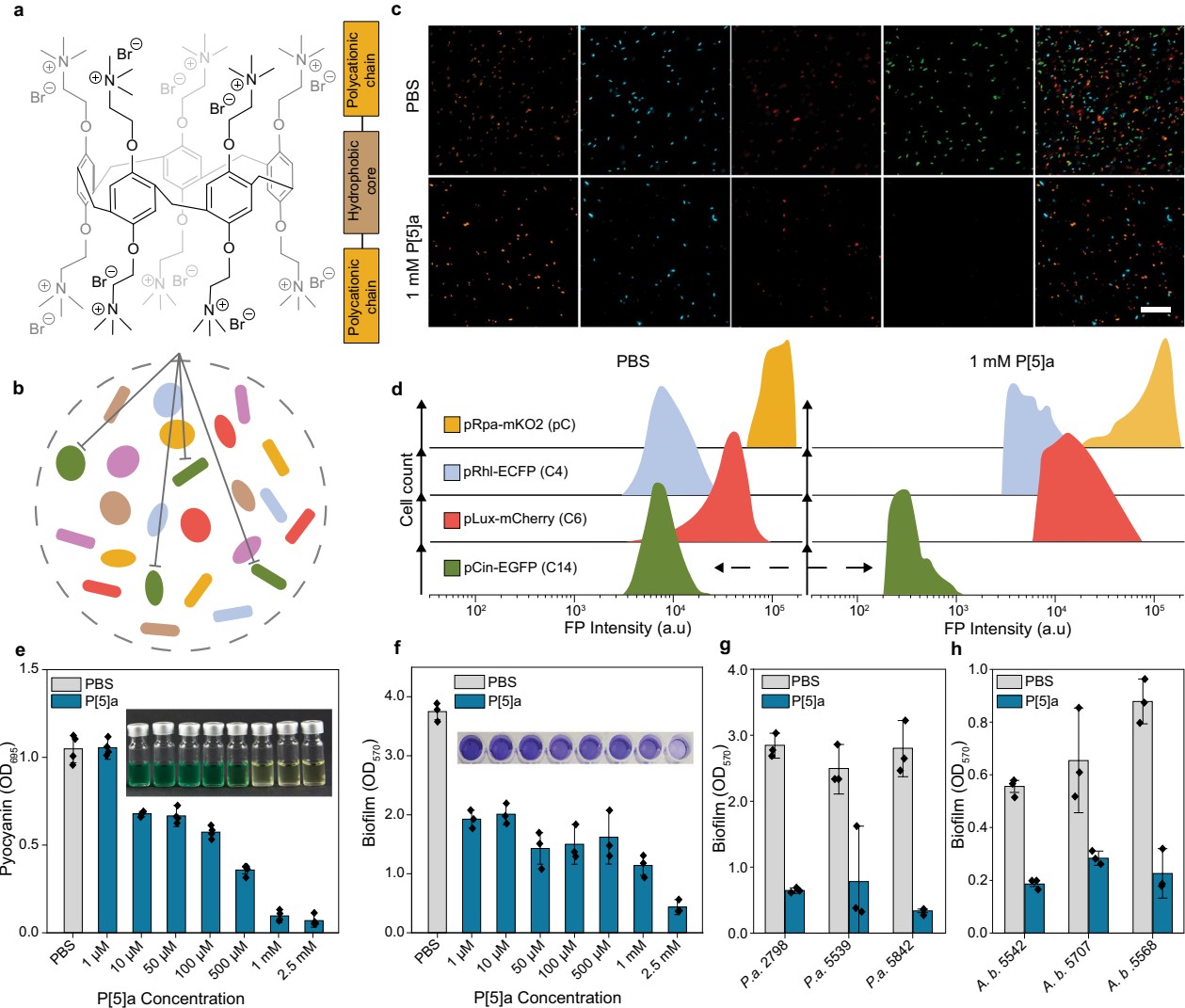

**Fig. 2 | P[5]a selective for prolonged acyl HSLs of Pseudomonas aeruginosa and Acinetobacter baumannii, resulting in an inhibition of biofilm and toxins.** **a** Chemical structure of the synthetic host P[5]a which demonstrated the strongest interaction with HSLs with longer acyl chains, highlighting the hydrophobic core cavity (brown box), flanked on both sides by the polycationic hydrophilic residues (orange boxes). **b** Schematic of a combined microbial community, where P[5]a specifically inhibits strains utilizing HSLs with longer acyl chains for communication and synchronization, as seen in (**c**) and (**d**). **c** Fluorescence microscopy images of a combined microbial community of E. coli cells consisting of four different strains with a unique fluorescent protein (pRhl-mKO2 orange for C4, pLux-CFP cyan for 3-oxo-C6, *pCin*-GFP green for 3-OH-C14:1 and pRpa-mCherry red for pC HSL) that each detect a different HSL, demonstrate that P[5]a specifically and preferentially interacts with HSLs that contain longer acyl chains. The addition of P[5]a results in the inhibition of the *pCin*-GFP, sensing the long chain 3-OH-C14:1 HSL.

Images representative of average values of biological replicates ± s.d ($n = 3$ per group). Scale bar, 50 μm. **d** Flow cytometry analysis on the combined microbial culture in (**i**), confirming the specific inhibition of *pCin*-GFP (sensing the 3-OH-C14:1 HSL) upon the addition of P[5]a, indicated by the black arrow. Offset in y-axis added for clarity purposes. **e** P[5]a suppresses the bacterial toxin pyocyanin, a key toxin in P. aeruginosa controlled by HSL quorum sensing, in PAO1 after a 24 h agitated culture. The data represent average values of biological replicates ± s.d ($n = 4$ per group). **f** P[5]a suppresses the formation of biofilm in PAO1 after a 24 h static culture. The data represent average values of biological replicates ± s.d ($n = 3$ per group). **g** P[5]a suppresses the biofilm formation in "superbug" MDR clinical isolates of P. aeruginosa after a 24 h static culture. The data represent average values of biological replicates ± s.d ($n = 3$ per group). **h** P[5]a suppresses biofilm formation in "superbug" MDR clinical isolates of *A. baumannii* after a 72 h static culture. Data represent average values of biological replicates ± s.d ($n = 3$ per group).

with the longer 3-Oxo-C12 and 3-OH-C14:1 HSLs, hence confirming complexation. Minimal changes were observed in C4 and pC HSLs in the presence of P[5]a, suggesting weak to no interaction (Fig. 3e, Supplementary Fig. 6).

We used density functional theory modelling (M06-2X)[32] to visualize the HSL-P[5]a interaction (Fig. 3f). The total binding surface areas were generally proportional to affinity. The large difference in alkyl chain volume between 3-oxo-C12 and 3-Oxo-C6 (70.5 vs 6.3 Å) correlated with their relative affinity. The 3-Oxo-C12 long hydrophobic chain resides in P[5]a's internal cavity, making this the most stable complex (Supplementary Fig. 7a, b). Furthermore, these HSL showed little to no disruption of the preferred geometry or electronics of the

host cavity, highlighting the complementarity (Fig. 3g, Supplementary Fig. 8). Notably, the presence of carboxyl, hydroxyl or absence of functional groups did not have an impactful effect on the binding affinities, apart from affecting the length of the acyl chain. This further indicates that binding affinities are largely determined by length and hydrophobicity of the acyl chain length.

## mRNA analysis reveals QS target and interplay of effects OM and extracellular space

Next we analysed the differential transcription of total mRNA in P[5]a treated and untreated PAO1 to gain more insight into the mode of action (Fig. 3h). Gene set enrichment using Kyoto Encyclopaedia of

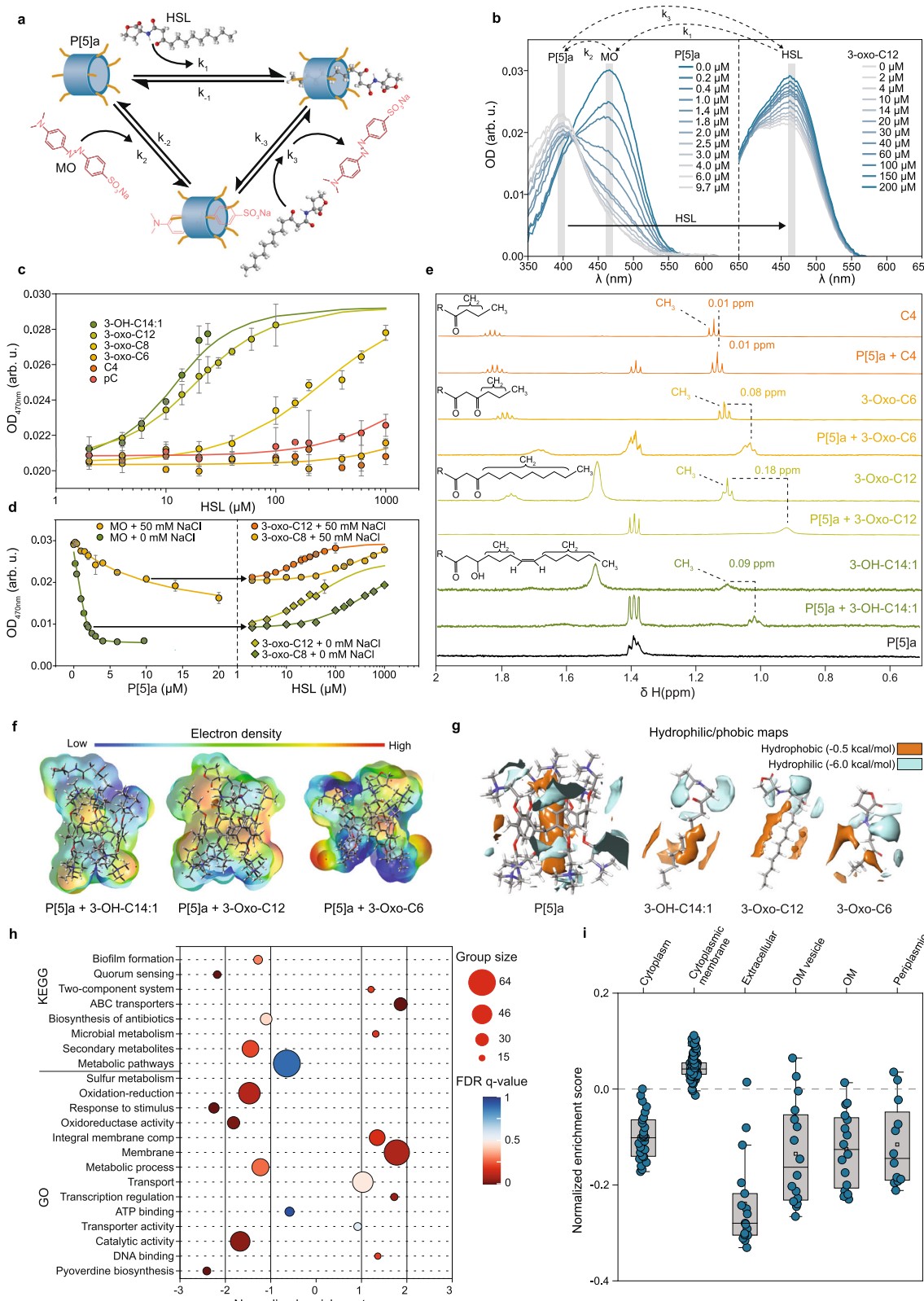

Genes and Genomes (KEGG) and Gene Ontology (GO) terms revealed strong down-regulation of virulence-associated factors, including biofilm formation, QS and pyoverdine biosynthesis. Localization of the protein products shows the largest downregulation in products targeted to the extracellular space and OM vesicles, both of which are strongly associated with virulent factors in *P. aeruginosa*, including LPS, exotoxins, proteases and elastases, and essential components in

the early stages of infections and their progression (Fig. 3i)[33]. We also observed up-regulation in two-component systems, integral membrane components and membrane gene sets, indicating an interplay of effects caused by P[5]a at the OM surface. This is in line with previous characterizations of cationic compounds and peptides, where the positively charged residues are speculated to interact with negatively charged lipids[34,35].

**Fig. 3 | Binding of HSLs occurs inside the cavity of the P[5]a macrocycle, with length of the prolonged acyl chain of the HSL playing a key role in the strength of the interaction. a** Schematic representation of the dye-displacement assay, showing the targeted P[5]a-HSL interaction ($k_1$), the P[5]a-MO complex formation ($k_2$) and the measured dye displacement assay ($k_3$). **b** Left: shift in absorption spectra of MO (2.0 μM) titrated with increasing amounts of P[5]a (*i.e.* $k_2$). Right: the titration of increasing amounts of 3-oxo-C12 HSL, displaces the MO inside the cavity of P[5]a, resulting in a recovery of the absorption spectrum, (i.e. $k_3$). **c** Measured optical density at 470 nm of P[5]a-MO complexes (P[5]a concentration = 10 μM, MO concentration = 2.0 μM, NaCl concentration=50 mM), titrated against increasing concentrations of the corresponding HSL (dots). The data represent average values ± s.d (*n* = 3 biologically independent samples per group). **d** Left: absorption spectra of MO (2.0 μM) titrated with increasing amounts of P[5]a ($k_2$) in 0 and 50 mM of NaCl. Right: absorption spectra of the P[5]a-MO complex titrated with the HSLs 3-oxo-C8 and 3-oxo-C12 ($k_3$) in 0 and 50 mM of NaCl. The data represent average values ± s.d (*n* = 3 biologically independent samples per group). Arrows indicate the P[5]a/MO ratio selected for each HSL titration (1:1 and 5:1, respectively). **e** NMR spectra (D₂O/[D₆]DMSO 30 % at 310 K) of P[5]a, the HSLs 3-OH-C14:1, 3-Oxo-C12, 3-Oxo-C6 and C4, and P[5]a combined with each HSL. **f** The electron density of the combined surface of P[5]a complexed with 3-OH-C14:1, 3-Oxo-C12, and 3-Oxo-C6 HSLs. **g** Hydrophobic/philic maps highlight the shape and extent of hydrophilic and hydrophobic surfaces of P[5]a in isolation, as well as in 3-OH-C14:1, 3-Oxo-C12 and 3-Oxo-C6. **h** Gene Set Enrichment Analysis (GSEA) against KEGG and GO gene sets, performed on sequenced mRNA of PAO1 treated with P[5]a. **i** Localization of mRNA products. Number of mRNA products per localization group ranges from cytoplasm to periplasm 27, 93, 17, 16, 16 and 12 respectively. Box plots show the median (centre line), the upper and lower quantiles (box), and the range of the data (whiskers).

## Table 1 | Binding affinities for host-guest interactions between the macrocycle P[5]a and HSLs, determined through dye displacement with the MO indicator

| HSL | $k_D$ (M) in 0 mM NaCl | $k_D$ (M) in 50 mM NaCl |
|---|---|---|
| $k_{-1}$ MO-P[5]a | (0.68 ± 0.08) nM | (9.82 ± 0.06) μM |
| $k_{-2}$ 3-OH-C14:1 (*Cin*) | (2.00 ± 0.20) μM [a] | (2.58 ± 0.07) μM [a] |
| $k_{-2}$ 3-oxo-C12 (*Las*) | (1.69 ± 0.05) μM | (5.90 ± 0.02) μM |
| $k_{-2}$ 3-oxo-C8 (*Tra*) | (18.60 ± 0.60) μM | (0.12 1 ± 0.02) mM |
| $k_{-2}$ 3-oxo-C6 (*Lux*) | (0.83 ± 0.04) mM | (1.36 ± 0.03) mM |
| $k_{-2}$ pC (*Rpa*) | (0.65 ± 0.05) mM | (4.36 ± 0.09) mM |
| $k_{-2}$ C4 (*Rhl*) | (15.00 ± 0.60) mM | – [b] |

[a] 3OH-C14:1 dataset is limited due to strong scattering. [b] Signal increase is insignificant to be fitted

### LPS sequestration is driven by electrostatic interactions

Given the effects of P[5]a on the bacterial membrane (Fig. 3h, i) and its net positive charge, we hypothesized a possible interaction with LPS. LPS is a key component in the bacterial OM whose phosphate groups contribute to the overall negative charge of the membrane surface. Dynamic light scattering (DLS) shows that the titration of purified LPS with increasing concentrations of P[5]a in water results in the formation of larger complexes, starting at 15 μM and plateauing at 40 μM (Fig. 4a). The titration of these complexes in water with NaCl reverses and breaks up the larger complexes, confirming that surface charge interactions are a driving force in the complexation (Fig. 4b, c)[36]. However, since most physiological conditions occur in relative high levels of salt, which hinders surface charge interactions, we explored the titration of LPS with P[5]a in physiological salt buffers 0.5x and 1x PBS. Starting at 50 μM in 0.5x and 1x PBS, we observed a continuous increase in the presence of large complexes. We further studied the P[5]a-LPS interaction in high salt (1x PBS) environments using analytical ultracentrifugation (AUC), to better understand the formed complexes (Fig. 4d). P[5]a and LPS separately required high rotational speeds of 60,000 RPM to achieve detectable sedimentation. P[5]a-LPS mixtures resulted in rapid sedimentation at rates as low as 25,000 RPM, suggesting the formation of larger complexes/aggregates (Supplementary Fig. 9a–c). P[5]a-LPS mixtures showed noticeable differences in the sedimentation speed of these complexes, as observed from the sedimentation coefficient range of 3–100 S. This range corresponds to molecular weights ranging from $5 \times 10^3$ up to $1 \times 10^6$ Da.

### P[5]a enhances the penetration of the OM of intracellular antibiotics

In order to explore whether the observed complexation with LPS affects its function in the outer membrane, we examined clinical isolates with reduced susceptibility to antibiotics (Supplementary Data 1). We co-administered P[5]a with four standard-of-care antibiotics with intracellular targets i.e. amikacin, cefepime, ceftazidime, and meropenem (Fig. 4e). In all combinations with P[5]a, minimum inhibitory concentration (MIC) values for the clinical isolates were reduced compared to the antibiotic treatment alone. The stronger reductions were not specific to a clinical isolate strain or type of antibiotic, indicating that the membrane interference enhances the penetration of all antibiotics with intracellular targets. This observation is in line with other OM-sensitizers[34,38].

We then studied the build-up of resistance in PAO1 towards antibiotic therapy over a 14-day period. Whereas in stand-alone antibiotic therapies of aztreonam, cefepime, meropenem and tobramycin (Fig. 4f-i), in all administrations encountered resistance, in the combined administration, no antibiotic encountered resistance. Aztreonam, cefepime and tobramycin remained fully susceptible, whereas meropenem reached intermediate susceptible. The average time for resistance levels to double was greatly increased (Fig. 4j). Collectively, these results emphasize the potential of P[5]a to revitalize available antibiotic therapies and reduce resistant encounters (Fig. 4k).

Isolates from acute infections often produce elevated levels of endotoxins, proteases, siderophores and demonstrate increased motility[39]. In contrast, isolates from chronic infections often prioritize a mucoid phenotype, lowered doubling times and increased biofilm production levels. Added to this variation is a plethora of increasingly prevalent resistance mechanisms. Together, these traits have increased the threshold for novel treatments significantly[40]. Consequently, we tested the effectivity of P[5]a against a large subset of 72 *P. aeruginosa* isolates, consisting of International Antigenic Scheme (IATS) serotype collection (19 isolates), a CF chronic collection (20 isolates) and a set of multidrug-resistant (MDR) clinical isolates with ranging antibiotic resistance profiles (33 isolates) (Fig. 5a, Supplementary Data 1). P[5]a showed widespread activity, reducing biofilm levels in over 96%, with over 92% significance. However, in one isolate, the biofilm production was significantly upregulated. We did not observe differences in the effects of P[5]a on different *P. aeruginosa* collections, bacterial phenotype, antibiotic resistance profile or site of isolation. Of note, P[5]a also proved effective against Δ*Las*R mutants (+), which have lost the ability to sense the 3-oxo-C12 HSL, which is particularly prevalent in *P. aeruginosa* strains isolated from chronic CF infections[41]. Ranking the documented resistant antibiotic profiles (supplementary Table 1) against the function of P[5]a (Fig. 5b), shows that P[5]a remains effective against, and evades, most antibiotic resistance mechanisms.

Given the widespread activity against clinical isolates and reduced resistance development against antibiotics in combined administrations, we next tested the resistance development in vitro (Fig. 5c). PAO1 was repeatedly passaged in the presence of P[5]a, where pyocyanin levels remained suppressed throughout the 14-day study. Next, we re-treated all MIC₅₀ cultures of PAO1 that were repeatedly passaged with the antibiotics cefepime and meropenem (Fig. 4g, h, respectively). Even as PAO1 developed rapid tolerance and resistance to the antibiotics, P[5]a remained fully effective (cefepime, zoom-in left and meropenem, zoom-in right in Fig. 5c). Recent efforts have uncovered that antibiotic tolerance precedes the development of resistance to

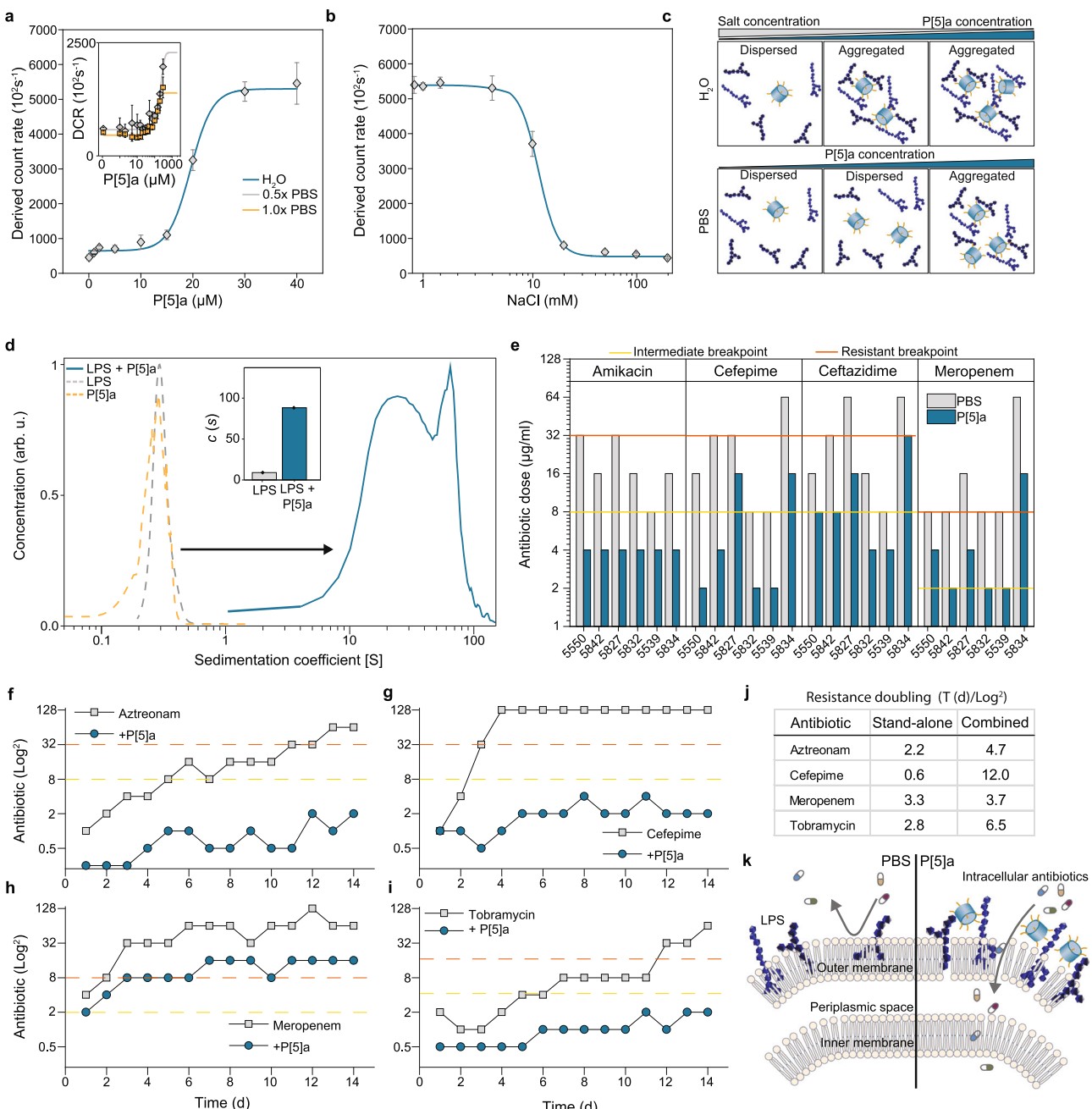

**Fig. 4 | P[5]a interacts with LPS in the bacterial OM, enhancing the penetration and potency of co-administered antibiotics. a** P[5]a titrated into 0.1 mg/ml PA10 LPS (5-10 μM) solution in water with DLS shows the formation and assembly of large complexes starting at 15 μM of P[5]a. The data represent average values ± s.d (*n* = 3 independent experiments per group). Expansion highlights the formation of P[5]a-LPS complexes in 0.5x and 1x physiological conditions. **b** The interaction and aggregation observed in (**a**) is reversible by the addition of salt (NaCl), which leads to the dissociation of the larger complexes. The data represent average values ± s.d (*n* = 3 independent experiments per group). **c** Schematic representation of the NaCl effects on the complexation of P[5]a-LPS. **d** Analytical ultracentrifugation shows the sedimentation coefficient of P[5]a alone on the left at 60,000 rpm and P[5]a with LPS of PA10 on the right at 25,000 rpm, revealing the presence of high-molecular weight structures, up to 1×10⁶ Da. Expansion highlights that a majority of the LPS, 91%, is present in the larger P[5]a-LPS complexes (*n* = 1 independent experiment per group). **e** P[5]a enhances the efficacy of co-administered antibiotics amikacin,

cefepime, ceftazidime, and meropenem in six MDR resistant *P. aeruginosa* clinical isolates. MIC values according to the Clinical and Laboratory Standards Institute (CLSI):[37] susceptible (below yellow line), intermediate susceptible (between yellow and red line), and resistant (above red line). **f–i** Serial passaging of PAO1 with the antibiotics aztreonam (**f**), cefepime (**g**), meropenem (**h**), and tobramycin (**i**) as well as co-administration of the antibiotics with P[5]a. Co-administrations greatly slowed the development of resistance of PAO1 to the respective antibiotic treatment. Cefepime reached 128 μg/ml after four days, which was the highest concentration of antibiotic included. **j** The average time (**d**) for PAO1 to double the MIC value required to kill the pathogen, as seen in (**f–i**), in both a stand-alone and combinatory treatment shows that co-administrations reduce the resistance doubling time. **k** Schematic representation of LPS positioned in the OM (left). P[5]a interacts with LPS in the OM, likely affecting the tight organization of the OM, which improves the effectivity of antibiotics with intracellular targets in a combined administration.

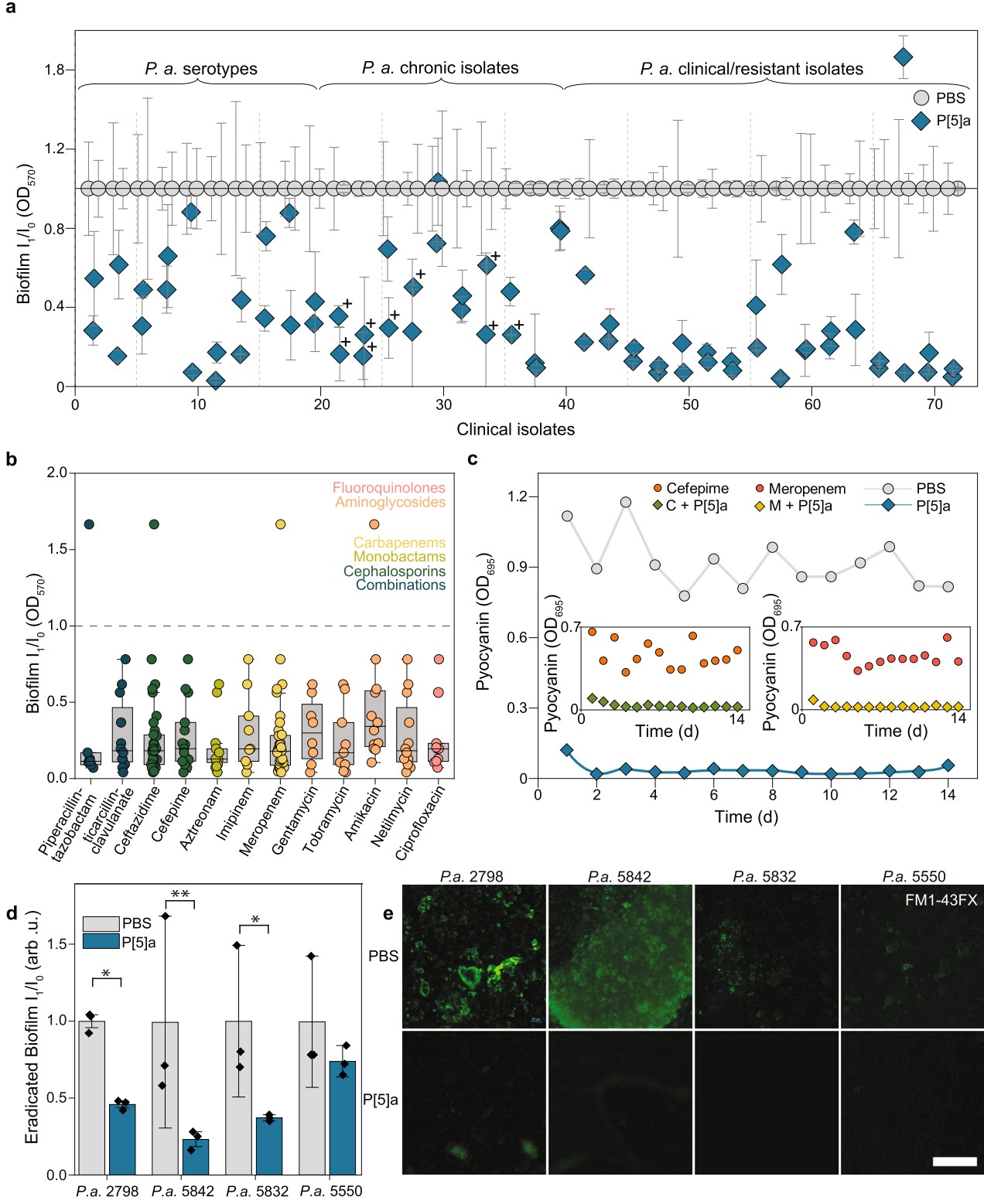

antibiotics[42,43]. Neither tolerance nor resistance mechanisms to antibiotics impacted the function of P[5]a in vitro, supporting the potential of non-bactericidal and/or bacteriostatic virulence-targeting strategies with reduced selective pressure[6,7,25].

Like the OM, the biofilm is generally negatively charged and mainly arises from LPS and extracellular polymeric substances[44,45]. We assessed whether P[5]a could also disrupt existing biofilms by exposing four highly resistant *P. aeruginosa* biofilms to P[5]a for 16 h after a

24 h formation period (Fig. 5d, e). Significant eradication of the biofilm was observed in all cases except *P. aeruginosa* isolate 5550. In *P. aeruginosa* isolate 5842, the biofilm mass was reduced by more than 75%.

## P[5]a protects A549 human lung cells against endotoxin and exotoxin-induced cell death
Biological membranes have striking similarities, with glycerophospholipids present in both eukaryotic bilayer membranes, as well as

**Fig. 5 | P[5]a enhances the penetration of co-administered antibiotics. a** Effect of P[5]a on biofilm inhibition after a 24 h static culture formation in a panel of *P. aeruginosa* isolates, consisting of *P. aeruginosa* IATS serotypes, a set of isolates from cystic fibrosis patients suffering from chronic infections, and a set of MDR clinical isolates, derived from various sites of infection. (+) Isolates with iridescent and metallic sheen of the colony surface, that is typical for a Δ*las*R mutant. The data represent average values of biological replicates ± s.d (*n* = 3 per group). **b** Effect of P[5]a on biofilm inhibition in marked resistant clinical isolates of panel seen in (**a**). P[5]a successfully inhibited biofilms in resistant isolates, independent of the type or mechanism of action of the antibiotic. The colours of the antibiotics correspond to the antibiotic class. Only isolates marked as resistant respective to antibiotic treatment were included. Number of resistant isolates per type of antibiotic ranges from Piperacillin-tazobactam to Ciprofloxacin 7, 12, 35, 15, 13, 9, 35, 8, 9, 12, 12 and 11 respectively. Box plots show the median (centre line), the upper and lower quantiles (box), and the range of the data (whiskers). **c)** Effect of P[5]a on pyocyanin toxin suppression over a 14 day period. P[5]a successfully suppressed toxin levels throughout the 14 days, with no observable decrease in effectivity. Expansion highlights the effects of P[5]a on toxin suppression of recultured MIC_{50} isolates from the cefepime (left) and meropenem (right) resistant stand-alone development experiments (Fig. 4g, h respectively). As tolerance, and subsequently resistance, develops rapidly to these antibiotics, P[5]a remains effective in suppressing toxins. **d** P[5]a disrupts biofilms formed by highly resistant clinical isolates of *P. aeruginosa* in vitro; mean biofilm mass remaining after 16 h of treatment with P[5]a. The data represent average values of biological replicates ± s.d (*n* = 4 per group). Two-tailed paired t-test **P < 0.01 *P < 0.05. **e** Fluorescence microscopy images of formed biofilms by *P. aeruginosa* isolates, demonstrate the eradication of formed biofilms by P[5]a after 16 h of treatment. Images are representative of three independent experiments. Scale bar, 200 μm.

the inner leaflet of the bacterial OM, and LPS present in the OM outer leaflet. We observed interactions between P[5]a and the OM, which resulted in permeabilization of the OM. So, we next turned to A549 epithelial cells with a bilayer membrane consisting of glycerophospholipids, major effector cells which constitute physical barriers against pathogens. We determined the effects of P[5]a on A549 epithelial cell viability using trypan blue staining, displaying low toxicity towards A549 cells and no observable membrane-lytic activity (Fig. 6a). This further suggests that the interactions with the bacterial OM occur through interactions in the membrane outer leaflet, where LPS is inserted, rather than impacting the inner leaflet consisting of glycerophospholipids.

We next checked whether P[5]a could protect epithelial cells against the full virulence arsenal of a clinically relevant pathogen. P[5]a protected A549 cells against apoptosis by PAO1 in a dose-dependent manner, with full protection at concentrations above 1 mM (Fig. 6b, c). The dose-dependence of P[5]a required to achieve complete protection in A549 against PAO1 was comparable to the amounts required to inhibit the virulence factors (pyocyanin, 3-oxo-C12, LPS, biofilm), suggesting that these virulence factors are the key cytotoxic agents utilized by the pathogen during infection.

To further understand the protective effects of P[5]a during bacterial infection, we analyzed total mRNA from the A549 epithelial cells (Fig. 6d). P[5]a alone had minimal effects on A549 epithelial cells, with none of the differentially expressed genes altered by more than log_{2}FC ± 1.2. Furthermore, we did not observe responses indicating changes in the stability of the bilayer membrane. The addition of PAO1 to A549 cells greatly altered expression levels, with gene set enrichment (GO Biological Processes) showing major changes in inflammatory response, response to molecules of bacterial origin, chemotaxis, cell death and regulation of cell proliferation (Fig. 6e, f). The biggest increase was observed in the chemokine ligands (CCL20, CXCL1, CXCL2, and CXCL3) together with the surface receptor (ICAM1), which all play key roles in immune and inflammatory responses[28]. The addition of P[5]a to infected A549 epithelial cells elicits a concentration-dependent reduction in the inflammatory response, chemotaxis, cell death, response to molecules of bacterial origin, and regulation of cell proliferation gene sets. The most significant dose-dependent reduction in expression was observed in chemokines CCL20, CXCL1, CXCL8 and ICAM1. We observed increased expression in adrenomedullin (ADM) and colony-stimulating factor 2 (CSF2), both implicated in the defensive response to bacterial infection[46,47], indicating that reduced stress and inflammatory responses increase the bacterial clearance capabilities.

## P[5]a sequesters LPS endotoxin in vitro and in vivo and protects against inflammatory responses

Recognizing the important role that virulence factors play during the establishment and spread of bacterial infections in their eukaryotic hosts, we further investigated the direct interactions between P[5]a and purified 3-oxo-C12 and LPS, and the subsequent effects on A549 epithelial cell viability. In bacteria, the 3-oxo-C12 HSL freely diffuses through bacterial membranes where it is utilized as a signalling molecule. However, in mammalian cells, the 3-oxo-C12 HSL interacts with bilayer membranes and results in the disruption of lipid domains. This results in the release of pro-inflammatory cytokines, TNFR1 signalling lymphocyte cell death, and an exacerbation of airway inflammation[48]. Similarly, LPS is a well-known pathogen-associated molecular pattern of which the Lipid A region is recognized by the toll-like receptor TLR4, resulting in the release of inflammatory cytokines, cell death responses, increased airway barrier permeability, and lung inflammation. A549 cells that were treated with purified 3-oxo-C12 or LPS showed an observable increased cell death at the first time point of 2 h, increasing steadily over time. The addition of P[5]a prevented epithelial cell death, both with 3-oxo-C12 and LPS, protecting the A549 cells over a 24 h period (Fig. 7a, b). These results show that i) the interaction rapidly impairs the ability of 3-oxo-C12 and LPS to disrupt lipid bilayer domains ii) the interaction is stable over time (over a 24 h period) and iii) the formed complexes do not dissociate.

We next investigated whether P[5]a protects against lung inflammation in female C57BL/6 J mice challenged with LPS. The intranasal administration of LPS, detected by the TRL4 receptor, results in the induction of a strong inflammatory response which is characterized by the infiltration of neutrophils and white blood cells into the extracellular matrix and by the release of cytokine that play a fundamental role in lung tissue damage (Fig. 7c). Three increasing concentrations of P[5]a (0.1, 0.5 & 3 mg/kg) were administered intratracheally (i.t.). At 24 h, 3 mg/kg P[5]a resulted in a 30% total reduction of LPS-stimulated white blood cells (WBC) and neutrophils in bronchoalveolar fluid (BALF), although not statistically significant (Fig. 7d, e, respectively). At 4 h, and at 0.1, 0.5 mg/kg dosages of P[5]a, no observable differences were seen. Next, we looked at the release of inflammatory cytokines in response to LPS stimulation. At 4 h, apart from IL-6 no significant differences were observed in cytokine levels between LPS stimulated and P[5]a treated groups (Supplementary Fig. 11). However, at 24 h, significant reductions were observed in cytokine levels upon the treatment of P[5]a. Importantly, TNF-α levels, a signature pro-inflammatory cytokine responsible for immune cell regulation and cell apoptosis, were significantly reduced even at the lowest dose of 0.1 mg/kg (Fig. 7f, Supplementary Fig. 12, 13,). Also the cytokine IL-6 and chemokines CXCL1 and CXCL2, key chemoattractants for macrophages and neutrophils, showed statistically significant reductions at the lowest dose of 0.1 mg/kg (Fig. 7g, h, i). Overall, these results show that P[5]a not only prevented epithelial cell damage, cell death and increased defensive responses, it also reduced their virulence-induced inflammatory activation. The latter is of major importance in vivo, because uncontrolled inflammatory responses lead to pulmonary exacerbations and extensive tissue damage[49].

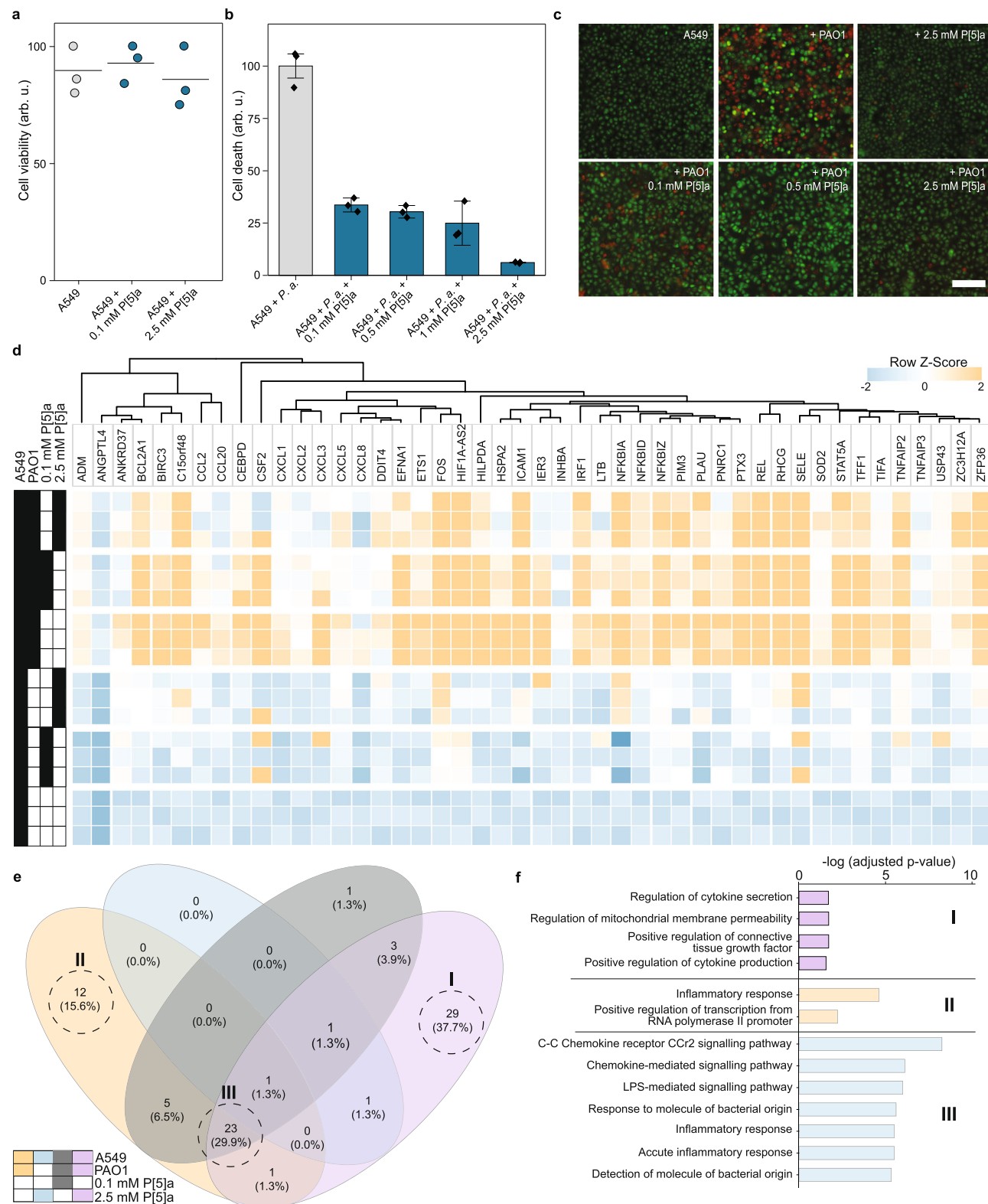

## Discussion

Antibiotic therapy is the mainstay of treatment against bacterial infection. However, their large-scale use combined with a limited spectrum of bacterial targets has fuelled resistant encounters, putting pressure on both existing therapies as well as the development pipeline.

Herein, we utilize an anti-virulence screen of host-guest interacting macrocycles, and identify a water-soluble synthetic Pillararene, P[5]

a (Supplementary Fig. 14–17), that is non-bactericidal/bacteriostatic. It presents a mechanism of action that involves binding to HSLs (inner cavity) and LPS (cationic rim), key virulence factors in Gram-negative pathogens.

We demonstrated that P[5]a effectively binds HSLs with different chain lengths with binding constants in the micromolar ranges. We observed an improved binding as the HSL guest alkyl chain length increases, suggesting that hydrophobic interactions contribute to the

**Fig. 6 | P[5]a protects A549 human epithelial cells against TNFR-1 dependent apoptosis by _P. aeruginosa_ infection. a** Trypan blue viability assessment of A549 epithelial lung cells treated with varying concentrations of P[5]a. The data represent average values of biological replicates. Mean (± SEM) (_n_ = 3 per group). **b** A549 epithelial cells challenged with PAO1 for 16 h show apoptosis. P[5]a recovers PAO1 induced apoptosis in a concentration-dependent manner. The data represent average values of biological replicates ± s.d (_n_ = 3 per group). **c** Fluorescent microscopy images of A549 epithelial cells as seen in (**b**), either untreated, P[5]a-treated, or challenged with PAO1 for 16 h. A concentration-dependent recovery of cell death is seen when treated with P[5]a. Scale bar, 50 μm. **d** Normalized expression values obtained from mRNA sequencing show that the addition of 0.1 mM and

2.5 mM is well-tolerated with no significant changes in gene regulation. A549 cells challenged by the bacterial pathogen PAO1 show strong activation of cellular stress, inflammatory and cell death responses. The addition of P[5]a treatment to the challenged A549 cells results in a concentration-dependent reduction in cellular stress, inflammatory and cell death responses, and also increases the defensive responses against bacteria (_n_ = 3); Two-tailed Wald test _p_ value, 0.01. **e** VENN-diagram of the gene regulatory network overlaps affected by infection and treatment. **f** Log₂FC expression changes related to differentially expressed GO Biological Process gene sets; I, II, II corresponds to diagrams in (**e**); Two-tailed BH adjusted p-value.

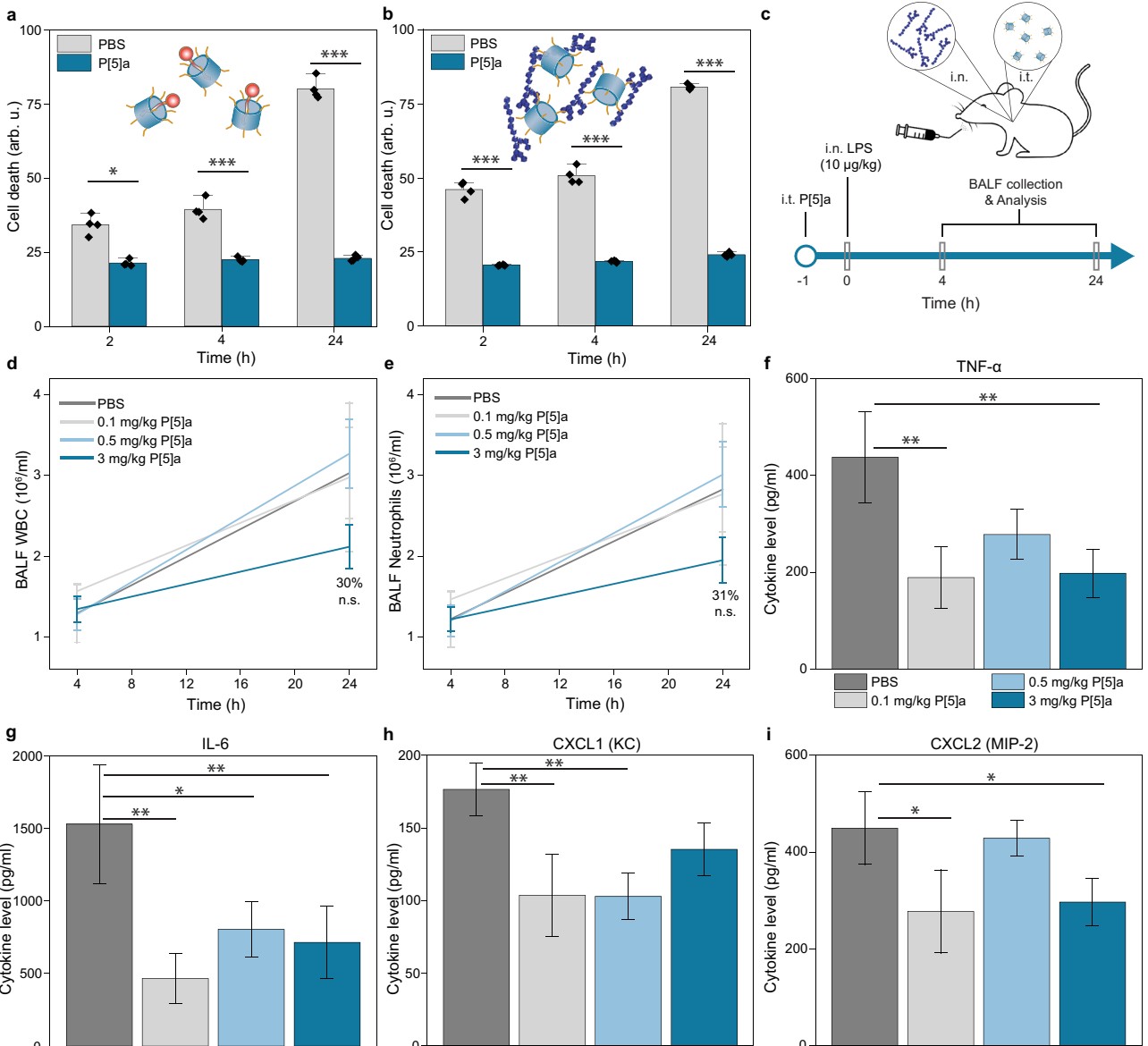

**Fig. 7 | P[5]a sequesters toxins and protects against LPS endotoxin induced inflammatory responses in A549 epithelial cells and in C57BL/J6 mice. a** The addition of P[5]a sequesters the 3-oxo-C12 HSL, protecting A549 epithelial cells against apoptosis over a 24 h period. The data represent average values of biological replicates ± s.d (_n_ = 4 per group). Two-tailed paired t-test \*\*\*_P_ < 0.001; \*\*_P_ < 0.01; \*_P_ < 0.05. **b** The addition of P[5]a protects A549 epithelial cells against LPS-induced apoptosis over a 24 h period(_P.a._ serotype 10, 20 μg/ml). The data represent average values of biological replicates ± s.d (_n_ = 4 per group). Two-tailed paired t-test \*\*\*_P_ < 0.001; \*\*_P_ < 0.01; \*_P_ < 0.05. **c**) LPS endotoxin sequestration study design. P[5]a is administered intratracheally one hour prior to the addition of

challenging agent, LPS (_P.a._ serotype 10, 10 μg/kg). BALF fluid is collected after 4 and 24 h and quantity of WBCs and neutrophils in BALF is quantified, along with inflammatory cytokines. **d**) Effects of P[5]a, dosed i.t., on LPS-stimulated total WBC counts in BALF. **e** Effects of P[5]a, dosed i.t., on LPS-stimulated total neutrophil counts in BALF. **f-i** Effects of P[5]a, dosed i.t., on LPS-stimulated (**f**) TNF-α, (**g**) IL-&, (**h**) CXCL1 (KC) and (**i**) CXCL2 (MIP-2) counts in BALF. One animal (3 mg/kg, 4 h) died immediately after i.t. treatments. Statistical analysis was performed by One Way ANOVA followed by Dunnett's test for multiple comparison, and Grubb's test was performed to exclude outliers. All groups vs veh/LPS treated group, \*_p_ < 0.05, \*\*_p_ < 0.01, Mean (± SEM) (_n_ = 10 animals per group).

recognition process. This view is both quantitatively and qualitatively supported by NMR, dye displacement and modelling of hydrophobic/hydrophilic maps, showing that the longer hydrophobic alkyl chains are placed deep into the aromatic cavity of P[5]a. The addition of NaCl did not have a large influence on the binding constants of the HSLs (independent of the length of the alkyl chains), suggesting that electrostatic interactions do not contribute to the recognition process and binding stability. The binding of the HSLs resulted in the inhibition of QS-associated virulence responses, including toxins, motility and biofilms. Combined, these results suggest P[5]a could function as an effective anti-virulence strategy, particularly in pathogens that utilize HSLs with long hydrophobic acyl chains.

On the other hand, we observed that electrostatic interactions play a key role in the binding of P[5]a to LPS on the OM, a highly impermeable barrier to otherwise effective antibiotic treatments. LPS on the OM is arranged and stabilized by divalent cations that form electrostatic bridges between the Lipid A domain[4], mainly $Ca^{2+}$ and $Mg^{2+}$. We hypothesize that the binding of P[5]a is mediated by the positively charged residues on the functionalized rim of P[5]a, which interact with the anionic Lipid A moiety. We observed that the interaction with LPS results in disruption of the OM organization, facilitating the access and potency of standard-of-care antibiotics to its intracellular targets, independent of their MoA (Fig. 4e). We also show that this synergy could be exploited to reduce resistance development in vitro and potentially revitalize antibiotic therapy (Fig. 4e–i). Supported by previous reports of cationic macrocycles[12,21] and the rational that LPS is a major component of bacterial biofilms[45], we investigated the eradication of formed biofilms by P[5]a. However, given the complexity of the composition of bacterial biofilms, future efforts are required to understand in depth the interactions that govern the eradication.

LPS is also the target of the antibiotic polymyxin B (PMB). Polymyxins competitively displace cations from the Lipid A domains, resulting in destabilization of the OM. However, where PMB also induces the depolarization or destabilization of the cytoplasmic membrane, bacterial viability assessments (Fig. 1f) showed that P[5]a did not have any observable effects on the cytoplasmic membrane. We speculate that the effects of P[5]a on the bacterial membrane might be more comparable to SPR741[34] and the OM-targeting antibiotics (OMPTAs)[50–52]. These molecules retain the ability to disrupt the outer leaflet of the bacterial OM, where LPS inserted, but lack the capability to interact with the bacterial OM inner leaflet consisting of glycerophospholipids and the cytoplasmic membranes. This is further supported the absence of lytic effects of P[5]a on eukaryotic lipid bilayer membranes, which are also composed of glycerophospholipids (Fig. 5a–e, supplementary Fig. 11).

Validation against MDR clinical isolates shows that P[5]a remains active against pathogens with an extensive variety of antibiotic resistance mechanisms, including carbapenem-resistant top priority *A. baumannii* and *P. aeruginosa*. Furthermore, P[5]a has no bactericidal/bacteriostatic effects, making the development of rapid resistance unlikely[53]. This is supported by the effective suppression of toxins throughout a 14-day period. In addition, as tolerance, and later resistance, developed to the antibiotics cefepime and meropenem over a 14-day treatment, at each timepoint, these isolates remained fully susceptible to P[5]a. Altogether, these results highlight the potential impact that differing bacterial targets and strategies can have over standard-of-care antibiotics, evading antibiotic resistance mechanisms and avoiding bacterial resistance build-up mechanisms; potentially circumventing some of the key bottlenecks currently faced in anti-infective development. A key question that remains is whether clinical isolates with resistance to colistin and PMB, of which 4-aminoarabinose and phosphoethanolamine modifications of the Lipid A are the primary mechanisms of resistance, affect the activity of P[5]a[54]. Interestingly however, P[5]a remained active against ΔLasR

isolates, which lack the 3-oxo-C12 HSL receptor (Fig. 5a). This suggests that the compound's polypharmacology could be a potent strategy to overcome partial resistance. As such, the dual mechanism of action of P[5]a previously reported[18], has in this work been exploited to target both the 3-oxo-C12 HSL (inner cavity, hydrophobic interaction) and LPS (outer polycationic rim, electrostatic interactions), and provides a promising approach to tackle bacterial resistance.

We observed that binding of P[5]a to both 3-oxo-C12 and LPS results in direct-acting sequestration of their toxic properties, both in vitro and in vivo. Toxins collectively disrupt tissue barriers, impede the host's immune defences against pathogens, and help promote bacterial growth and colonization and increase chances of successful infection[7]. 3-oxo-C12 and LPS challenged A549 cells were fully protected upon the addition of P[5]a (Fig. 5a, b). The binding of LPS also further supports the specific interaction of P[5]a with the Lipid A moiety, since specifically the Lipid A-region is recognized by the TLR4 receptor that results in cytokine release and cell death responses. Furthermore, the addition of P[5]a to A549 cells challenged with PAO1, showed a dose-dependent recovery of cell death, indicating that toxins and other virulence factors play a central role in cell death responses and the spreading of infections, which is consistent with previous observations (Fig. 5c–g)[6,7].

Our data provides a compelling argument for pursuing macrocycle-based virulence inhibitors as a paradigm for the treatment of bacterial infections. Their dual functionality of inhibiting virulence through HSL sequestration in their inner cavity, together with OM potentiating effects that sensitize pathogens to antibiotics, supports the use of P[5]a as a stand-alone as well as combinatory treatment with existing anti-infectives. Our findings also present a promising outlook on the use of rational design host-guest chemistry towards the synthesis of dual-acting molecules that can be used in the treatment of antibiotic-resistant bacterial pathogens.

## Methods

### Ethical statement
Our research complies with all relevant ethical regulations; Italian Legislative Decree No. 26/2014 and European Directive No. 2010/63/UE. The study was conducted in accordance with national legislation, under approval of the internal Aptuit Committee on Animal Research and Ethics and under authorisation issued by the Italian Ministry of Health (Italian Ministry of Health Authorisation Project – Internal Code No. 30004-B64).

### Cell lines, bacterial strains & growth conditions
A full table of cell lines, bacterial strains, and plasmids used in this study are listed in Supplementary Table 1. *E. coli* strain Top10 was used for cloning and plasmid construction, while strain CY008 (BW25113 ΔlacI ΔaraC ΔsdiA; Bennett lab, Rice University, Texas, US) was used for all fluorescence measurements.

Individual cultures were grown overnight in Luria Broth (LB) with appropriate antibiotics (ampicillin, 100 µg mL$^{-1}$, and kanamycin, 50 µg mL$^{-1}$) at 37 °C.

*P. aeruginosa* and *A. baumannii* clinical isolates were obtained from several sources: CF clinical isolate panel was obtained from Dr. A. Bragonzi, group leader of the Infections and Cystic Fibrosis Unit at the San Raffaele Institute.

Clinical isolates of *P. aeruginosa* and *A. baumannii* were obtained from the Helsinki University Hospital (HUS) collection, or isolated from patients by Chief Physician Dr. A. Pätäri-Sampo. The collection of the isolates fully complied with the Act on the Status and Rights of Patients" (785/1992). As such, information collected on the clinical from HUS isolates was limited and in no way traceable to individual patients and the information documented was restricted to the bacterial serotype, site of isolation and antibiotic resistance profile.

"Global Priority superbugs" were obtained from the ATCC collection of clinically relevant, drug-resistant strains with source metadata and genotypic and phenotypic characterization.

A549 human lung carcinoma cells were treated with TrypLE (Thermo Fisher Scientific) prior to the start of the experiment, split, counted, and aliquoted into 96-well plates at 20,000 mammalian cells per well. The cell counts were estimated using a TC20 cell counter (BioRad). Cells were grown for 24 h at 37 °C in DMEM (Gibco) with 10% fetal bovine serum (Gibco), 1× PenStep (Sigma), and 1× GlutaMAX (Gibco). After the 24 h growth period, cells were washed with warm 1x PBS.

### Cloning
Plasmids were constructed as previously described[55]. Briefly, fragments were cloned into two different vector backbones, pACYC (amp) for the receptor proteins and pColE1 (kan) for the AHL promoters, which were fused to the fluorescent proteins. All sequences were verified through DNA sequencing. Fluorescent proteins ECFP, EGFP, mKO2, and mCherry were tagged with the LAA degradation tag sequence. Plasmids are described in Supplementary Information Table 2.

### Materials
Homoserine lactones (HSLs; *N*-Butyryl-L-HSL (C4), *N*-(ß-Ketocaproyl)-DL-HSL (> 98%) (3-oxo-C6), *N*-(p-Coumaroyl)-L-HSL (> 94%) (pC), N-(3-Oxoocatnoyl)-L-HSL (> 97%) (3-Oxo-C8) and *N*-(3-Oxododecanoyl)-L-HSL (> 98%) (3-oxo-C12) were used as received from Sigma-Aldrich. *N*-(3-Hydroxy-7-cis-tetradecanoyl)- L-HSL (> 95%) (3-OH-C14:1) was used as received from Cayman Chemical. Organic solvents were purchased from Sigma-Aldrich and VWR. Methyl orange was purchased from Sigma-Aldrich. Lipopolysaccharides, purified through phenol extraction from *Pseudomonas aeruginosa* 10, was purchased from Sigma-Aldrich. 12-crown-4, 2-hydroxymethyl-12-crown-4, 1-Aza-15-Crown-5, 15-Crown-5, 4-sulfocalix[4]arene, 18-Crown-6, Cucurbit[6]uril hydrate, (2-hydroxypropyl)-α-cyclodextrin, α-cyclodextrin, Calix[6]arene, 4-tert-butylcalix[6]arene, Methyl-β-cyclodextrin, (2-hydroxypropyl)-β-cyclodextrin, β-cyclodextrin, 2-hydroxypropyl-γ-cyclodextrin, γ-cyclodextrin and Calix[8]arene were purchased from Sigma Aldrich & Merck. The P[5]s was synthesized according to reported procedures with full characterization[56,57]. Briefly, 1,4-bis(2-hydroxyethoxy)benzene was converted to 1,4-bis(2-bromoethoxy)benzene in the presence of carbon tetrabromide and triphenylphosphine in acetonitrile. In the next step, the decabromopillar[5]arene was obtained from the reaction of reacting 1,4-bis(2-bromoethoxy)benzene and paraformaldehyde in the presence of boron trifluoride diethyl etherate in dichloromethane under argon atmosphere. In the final step, the decabromopillar[5]arene is refluxed with trimethylamine in ethanol, with the final P[5]a collected as a precipitate after washing with more ethanol. The Calix[4]resorcinarene was previously synthesized by us with full characterization therein[58].

### Biofilm Inhibition assay
A total volume of 200 μL was prepared for each sample, and measured in a 96-well plate (Corning) at 20 °C. Each sample was blank subtracted and zeroed at 570 nm using a Cytation 3 (AHDiagnostics) or Hidex Sense (Hidex) plate reader. The samples were prepared in triplicates, and the average and standard deviation shown. *P. aeruginosa* was cultured overnight in LB medium containing 1% D-Glucose, and *A. baumannii* - in LS-LB (0.5% NaCl, 0.1% tryptone, 0.5% yeast extract) + 1% D-glucose Overnight *P. aeruginosa* and *A. baumannii* cultures were diluted 1:100 in a transparent 96-well plate with 200 μL LB medium containing 1% D-Glucose. A concentration of 1 mM P[5]a was added from a 100× stock or equivalent concentration of distilled H$_2$O. Cultures were incubated for 24 h at 37 °C. The supernatant was then removed, and wells were washed three times with sterile 1× PBS, once

with ice-cold methanol to fix the biofilm, and left to dry. A 0.1% crystal violet solution (Sigma-Aldrich, > 90%) was added to stain the biofilm for 15 min, followed by three washing steps with distilled H$_2$O to remove unbound dye. 96% EtOH was added to solubilize the biofilm, left for 30 min under mild shaking. The solution was then transferred to a new sterile 96-well plate, and OD$_{570}$ was measured.

### Dye displacement assay, fitting and binding constant determinations
A total volume of 200 μL was prepared for each sample, and measured in a 96-well plate (Corning) at 20 °C. Each sample was blank subtracted, and zeroed at 650 nm using a Cytation 3 (AHDiagnostics) or Hidex Sense (Hidex) plate reader. The samples were prepared in triplicates, and the average and standard deviation shown. Binding constants were obtained by Reactlab Equilibria® software (Jplus consulting), fitting the averaged spectra of each sample between 390 and 490 nm. Dissociation constants were calculated as the inverse of the association constants. Full binding asymptotes were simulated by Reactlab EQSIM2 software (Jplus consulting) with the parameters obtained from fitting.

The MO-P5a binding constant determination (k$_2$): A constant concentration of MO ($2.0 \times 10^{-6}$ M) was titrated with increasing amounts of P5a, at 0 and 50 mM of NaCl (Fig. S1a, b). Both wavelengths of independent triplicates were successfully fitted to a 1:1 binding model (Fig. S1c).

The P5a-HSL binding constant determination (k$_1$): A concentration of MO-P5a complex was titrated with increasing amounts of HSLs. In order to ensure the complex formation, different conditions were used at 0 and 50 mM of NaCl.

- NaCl = 0 mM: [MO] = [P5a] = $2.0 \times 10^{-6}$ M
- NaCl = 50 mM: [MO] = $2.0 \times 10^{-6}$ M; [P5a] = $1.0 \times 10^{-5}$ M

Samples were prepared by adding a constant volume (2 μL) of varying concentration of HSLs in pure DMSO into 198 μL of the abovementioned solutions.

### Pyocyanin assay
A total volume of 200 μL was prepared for each sample, and measured in a 96-well plate (Corning) at 20 °C. Each sample was blank subtracted and zeroed at 695 nm using a Cytation 3 (AHDiagnostics) or Hidex Sense (Hidex) plate reader. The samples were prepared in quadruplicates, and the average and standard deviation shown. Overnight *PAO1* cultures were diluted 1:100 in 200 μL LB medium. Varying concentrations of P[5]a were added from a 100× stock or equivalent concentration of dH$_2$O. Cultures were incubated for 24 h at 37 °C. The culture fluid was obtained via centrifugation at $16.3 \times 1000\,g$ for 15 min and passed through 0.22 μm syringe-driven filters. The cell-free solution was transferred to a new sterile 96-well plate, and OD$_{695}$ was measured.

### Fluorescent *E. coli* reporter system
The fluorescent reporter system of 6 *E. coli* strains detecting 6 different HSLs (Fig. 1b) was developed in this study to screen macrocycles that affect bacterial signalling. To create the reporter strains, *E. coli* CY008 strain was transformed with two plasmids (Supplementary Table 2): one with a gene encoding a receptor protein unique to a certain HSL, under a constitutive promoter, and the second one has a gene for a fluorescent protein (EGFP, for the reporter system; and EGFP, mKO2, CFP, and mCherry for combined microbial communities) under an inducible promoter specific to the certain HSL. The externally added HSL binds to the receptor protein triggering its dimerization. Upon the dimerization, the receptor protein binds to the promoter and activates the fluorescent protein gene expression.

To test the macrocycle interference with HSLs, the overnight cultures of the each reporter *E. coli* strains were diluted 1:1000 and grown

until OD600 of 0.2 at 37°C with 220 rpm shaking. Then the cultures were transferred to the 96 well plate and mixed with the corresponding HSLs ($10^{-6}$ M) and 0–2.5 mM of macrocyclic compound. Macrocycles were dissolved in 1X PBS, unless otherwise indicated (Supplementary table 1, last column). For those macrocycles where DMSO was required for solubility, final concentrations in the assay were kept at 1%. The plate was incubated at 37°C with 180 rpm shaking. The absorbance OD600 and fluorescent signal development were monitored with Cytation 3 plate reader (BioTek). The stronger the binding between the macrocycle and HSL, the less free signalling molecule is present that can activate fluorescence expression in the *E. coli*.

The interactions of the macrocycles with the HSLs were detected by decrease of the ratio of the fluorescence intensities at 6 h with the macrocycle added ($I_1$) and without the macrocycle ($I_0$).

## Flow cytometry
The *E. coli* strains producing EGFP, mKO2, CFP, and mCherry in response to HSLs were analyzed at a low flow rate in a FACS™Aria III system (BD Biosciences, San Jose, CA) using 488 nm, 561 nm, 405 nm and 633 nm lasers, correspondingly. For each sample, 10,000 events were measured. Data for the CFP/DAPI-A (450-40), GFP/FITC-A (530-30), mKO2/PE-A (582-15) and mCherry/PE-Cy5-A (670-14) channels were collected as pulse area, and the compensation procedure was performed to avoid overlapping signal collection.

## Biofilm eradication assay
A total volume of 200 μL was prepared for each sample, and measured in a 8-well coverglass chambers (Nunc Lab-Tek) at 20°C. The samples were prepared in triplicates, and the average and standard deviation shown. Biofilms of the extensively resistant *P. aeruginosa* isolates 2978, 5550, 5832 and 5842 were established by culturing for 24 h in 8-well coverglass chambers (Nunc Lab-Tek) filled with LB medium containing 1% D-Glucose, starting from an overnight culture diluted 1:100. The medium was removed and subsequently treated with 45.2 μg P[5]a or PBS for an additional 16 h, as previously reported[59]. The biofilms were then washed with water, and FM1-43FX (Invitrogen) was added to stain the bacterial cell membrane. The biofilm was fixed in 16% paraformaldehyde, 2.5% glutaraldehyde and 4.0% acetic acid in 0.1 M phosphate buffer (pH 7.4) overnight at 4°C. Biofilms were visualized using an Axio Observer Z1 microscope (Carl Zeiss, Jena, Germany). Green fluorescent signal was obtained using excitation light at 470 nm, while collecting the emitted light of 515–535 nm.

## Viability assessment of A549 cells
A total volume of 200 μL was prepared for each sample and measured in a 96-well tissue culture treate plate (Corning). Each sample was blank subtracted and zeroed at 600 nm using a TC20 cell counter (BioRad). The samples were prepared in biological triplicates, and the average and standard deviation shown. The master mix contained 450 μl DMEM medium (10% FBS, 1x PenStrep, 1x Glutamax) and either 2.5 mM or 100 μM of P[5]a. Cells were incubated for 16 h at 37°C. After the experiment, the adhered cells were treated with TrypLE (Thermo Fisher Scientific) and mixed with free cells from the supernatant. The collected cells were mixed 1:1 with TrypanBlue (BioRad) and counted.

## Cytotoxicity assay
A549 cells were grown as described above. The samples were prepared in biological triplicates, and the average and standard deviation shown. Before the experiment, the cells were treated with TrypLE (Thermo Fisher Scientific), split, counted, and aliquoted into 96-well plates at 50,000 mammalian cells per well. Cells were grown for 16 h at 37°C in DMEM with the following additives: 10% fetal bovine serum (Gibco), 1× PenStep (Sigma) and 1× GlutaMAX (Gibco). The cells were then washed with 37°C PBS, and 50 μL of "master mix" was added to each well for each condition. The master mix contained 450 μl DMEM medium

without any additives, CellTox Green Dye (CellTox™ Green Cytotoxicity Assay, Promega), either 2.5 mM or 100 μM of P[5]a. The fluorescence was read with a Hidex Sense (Hidex) plate reader with 490 nm excitation source and a 520 nm emission filter. Cells were incubated for 16 h at 37°C. The final results were based on the difference in fluorescence between 2 h and 5 h measurement time points. The linear range and sensitivity of the experiment was determined using DMSO as the toxic agent.

## Mammalian cell line infection model
A549 cells were grown and prepared as described in "Cytotoxicity assay". Cells were either challenged with i) 10 μM of purified 3-oxo-C12 HSL, ii) 100 μg of purified LPS, or ii) 50 μl of OD600 = 2 *P. aeruginosa*. The fluorescence was read with a Hidex Sense (Hidex) plate reader with 490 nm excitation source and a 520 nm emission filter. Cells were incubated for 16 h at 37°C. The final results were based on the difference in fluorescence between 2 h and 5 h measurement time points. Phase contrast and fluorescent images were acquired using an Axio Observer Z1 microscope (Carl Zeiss, Jena, Germany). Green fluorescent signal was obtained using excitation light at 470 nm, while collecting the emitted light of 515–535 nm. Red fluorescent signal was obtained using excitation light at 590 nm, while collecting the emitted light of 615–675 nm.

## Analytical ultracentrifugation
Sedimentation velocity experiments were carried out in a Beckman Coulter Optima analytical ultracentrifuge. All experiments were performed at 20°C using standard 2-channel centerpieces in an An-60 Ti rotor. P[5]a and lipopolysaccharide from *Pseudomonas aeruginosa* 10 (Sigma-Aldrich) were measured at 75 μM and 0.5 g L$^{-1}$ concentrations, respectively. Measured mixture of P[5]a and lipopolysaccharide were obtained by mixing P[5]a and lipopolysaccharide in ratio 4:1 (final concentration in mixture: 125uM of P[5]a and 0.5 g L$^{-1}$ of lipopolysaccharide). 0.5 x PBS buffer was used for all samples. Sedimentation velocity experiments were performed in absorbance mode at 260, 290 and 305 nm wavelengths and at two rotor speeds: 25,000 and 60,000 RPM. Data analysis were performed by Sedfit and Ultrascan software.

## Dynamic light scattering
A total volume of 300 μL was prepared for each sample and measured in semi-micro PMMA cuvettes (Brandtech scientific) at 20°C. Each sample was blank subtracted, and zeroed. The hydrodynamic diameter (Dh) of the assemblies was measured using a Malvern Instruments DLS device (Zetasizer Nano ZS Series) with a 4 mW He-Ne ion laser at a wavelength of 633 nm and an avalanche photodiode detector at an angle of 173°. The samples were prepared in triplicates, and the average and standard deviation shown. 100 mg L$^{-1}$ of purified LPS from PA10 (phenol extracted) was dissolved in either 300 μL distilled $H_2O$, 0.5x or 1x PBS buffers and titrated with P[5]a (0.1–17.5 mg L$^{-1}$) to achieve the desired ratio. Because the volume addition did not exceed 5 % of the total sample volume, no dilution corrections were included. for the titration of LPS-P[5]a complexes (40 μM P[5]a), a 0.01–0.5 M solution of NaCl was used to disassemble the formed complexes.

## RNA-sequencing
The RNA from A549 cells and *P. aeruginosa* PAO1 was extracted using an RNeasy Mini Kit (Qiagen), prepared and sequenced using a method designed based on the Drop-seq protocol[60] and further described in (Supplementary methods Sequencing). The read alignment and RNA-seq data analysis are further described in (Supplementary methods Read alignment and RNA-seq data analysis).

## NMR Spectroscopy
$^1$H NMR spectra were recorded on a Bruker Avance DRX 400 spectrometer. 5 mM stock solutions of the receptors (4-Sc[4]a, P[5]a and

β-CD), the HSLs (C4, pC, 3-Oxo-C6, 3-Oxo-C12 and 3-OH-C14) were prepared. Pure receptors and HSLs were prepared to a 2.5 mM sample concentration by adding 250 μL of stock solution to 250 μL of pure solvent ($D_2O$ or LB). For 1:1 host-guest (receptor-HSL) mixtures, 250 μL of the host and 250 μL of the guests were measured for a 2.5 mM concentration of both the host and the guests. The spectra were calibrated using $D_2O$ signal as an internal standard.

## Theoretical and Computation Calculations

All structures were computed by applying the Gaussian 09 suite of programs[61] using the Minnesota M06-2X, a hybrid metaGGA[62] exchange-correlation functional together with the Pople's 6–311 G** basis set, and the integrated equation formalism polarized continuum solvation model (IEFPCM) to consider solvent effects. To further validate the selection of this basis set and level of theory for both long-ranged and short-ranged interactions within the systems, all optimizations were repeated using the long-range dispersion-corrected functional (ωB97X-D)[63,64] and the combined basis sets; LANL2DZ for Br, and 6–311 G** for the rest of atoms. This level of theory is reported to be appropriate for modelling the H-bonding and dispersive non-covalent host-guest interactions. All minima were confirmed by the presence of only real vibrational frequencies. Thermochemical quantities were evaluated at 298 K. Single-point calculations were also performed at M062X/6-311 + G** level on M062X/6-311 G** optimized geometries. Structure visualization was performed using GaussView v5.0.8.4 and Maestro 11[65]. In addition, the quantum theory of atom in the molecule (QTAIM) was applied for a topological analysis of the key interactions within the cavity of a host and their density characteristics using AIM2000 software. The charge distribution was analyzed using a molecular electrostatic potential surface (MESP) map.

To study the relationship between the structure and activity of different host-guest systems, the molecular electrostatic potential (MEP) and the MESP shift, which is the most useful electrostatic property, were investigated. Molecular surface electrostatic potential shifts for the host-guest complexes were calculated as a valuable recognition tool in host-guest assembly, in which the density cube of P[5]a, guest fragments and complexes were calculated to generate an isosurface. We then mapped the values of the ESP shift within the complex onto these isosurfaces. The molecular orbitals (HOMO and LUMO) of the complex due to the binding of different ligands (3-OH-C14, 3-Oxo-C12, 3-Oxo-C6) in the P[5]a binding pocket were computed at the M062X/6-311 G** level.

To further evaluate the key intermolecular interactions, QTAIM analysis was applied. A topological analysis of the electron density was carried out with Bader's quantum theory of atoms in molecules (QTAIM) using AIM2000 software.

## MIC determination

The MICs of active bacterial cultures in the presence of antibiotics and/or P[5]a were determined via the Mueller broth microdilution assay according to Clinical and Laboratory Standards Institute (CLSI) guidelines[37]. The assays were performed in 96-well polystyrene flat-bottom microtiter plates (NUNC). Briefly, a single colony was inoculated in 5 mL LB medium and grown to the end-exponential growth phase in a shaking incubator at 37 °C. Cultures were subsequently diluted to an OD600 (optical density) of 0.002 (equivalent $1 \times 10^8$ CFU mL$^{-1}$) in fresh LB medium. 100 μL of LB medium with different concentrations of antibiotics, and in P[5]a where applicable, were serially diluted in a sterile 96-well plate. Afterwards, 100 μL of the diluted bacteria were pipetted into 96-well plates. In each plate, the grown bacteria with the maximum concentration of carrier and medium were considered as positive and negative controls, respectively. The 96-well plates were then statically incubated overnight at 37 °C to allow bacterial growth.

## Multistep P[5]a resistance development study

The ability of a susceptible *P. aeruginosa* lab strain to develop resistance against P[5]a was assessed using an adaptation of a multistep resistance study by repeated daily subculturing in the presence of P[5]a, carried out over 14 days. Briefly, *P. aeruginosa* O1 cultures were grown overnight with 1 mM P[5]a in LB medium, then the OD of bacteria was adjusted to an OD600 of 0.002. Bacterial cells were cultured for 24 h under constant agitation. Afterwards, the culture fluid was obtained via centrifugation at $16.3 \times 1000\,g$ for 15 min. Culture fluids were passed through 0.22 μm syringe-driven filters. The cell-free solution was transferred to a new sterile 96-well plate, and $OD_{695}$ was measured using a Cytation 3 plate reader (AHDiagnostics). The cultures were used for a subsequent dilution with fresh P[5]a.

## Multistep antibiotic resistance development study

The ability of a susceptible *P. aeruginosa* lab strain to develop antibiotic resistance in the presence or absence of P[5]a was assessed using a multistep resistance study by repeated daily subculturing in the presence of the half-MIC value of the active antibiotic concentration. This was carried out over a period of 14 days according to CLSI guidelines. Briefly, *P. aeruginosa* O1 cultures were grown in Mueller broth medium, then the OD of bacteria was adjusted to an OD600 of 0.002. Bacterial cells were treated by the aggregator at half-MIC concentration; after a 24 h incubation period, the MICs were tested by a microdilution assay.

## In vivo sequestration of LPS in C57BL/6 J mice

This study was subject to legislation under the Italian Legislative Decree No. 26/2014 and European Directive No. 2010/63/UE. The study was conducted in accordance with national legislation, under approval of the internal Aptuit Committee on Animal Research and Ethics and under authorisation issued by the Italian Ministry of Health (Italian Ministry of Health Authorisation Project – Internal Code No. 30004-B64). General procedures for animal care and housing are in accordance with the current Association for Assessment and Accreditation of Laboratory Animal Care recommendations. Sex was considered in the study design. Only female mice were selected for this study, since male mice show greater fluctuations in body temperature and sickness behavior in response to immune challenges by lipopolysaccharides[66]. 8-week-old female C57BL/6 J mice (Charles River France) were used for the LPS-induced lung inflammation model.

- Intratracheal P[5]a administration
  All animals were anaesthetized (4.0–4.5% isoflurane; 2 l/min $O_2$) and placed on an angled dosing table placed within a fume hood. The animals were positioned on their backs and suspended on a wire by their top two teeth. The animals' tongues were gently pulled out to one side using forceps. A cannula connected to PE 100 tubing and a 100 μl Hamilton syringe were inserted into the trachea. 50 μl of vehicle or P[5]a were dosed into the lungs. The animals were maintained in an upright position for a few seconds before being returned to their home cage.

- Intranasal LPS challenge
  One hour after i.t. administration of vehicle or P[5]a, all animals were anaesthetized (4.0–4.5% isoflurane; 2 l/min $O_2$). Once an adequate level of anesthesia was reached, the animals were i.n. dosed with LPS (50 μl/mouse) using a P100 pipette (Gilson). The animals were held upright for a short period of time after dosing to allow for substance distribution down the respiratory tract. The animals were then placed into a recovery cage and, once fully conscious, were returned into their home cage.

- BALF collection and inflammatory cell counts
  Four or twenty-four hours after challenge, animals were sacrificed using a combination of isoflurane and an intraperitoneal injection of thiopental sodium (300 mg/kg). The trachea was exposed and a guiding cannula connected to a syringe was inserted. Lungs were

washed three times with 0.4 ml of PBS. The recovered BALF was centrifuged at 800 x g for 10 min at 4 °C. Pellets were re-suspended in 200 μl of PBS and cells were counted using an automatic cell counter (Dasit). BALF supernatant was aliquoted and stored at −80 °C for subsequent analysis of mediator levels.

– Analysis of inflammatory mediators in BAL fluid.

The quantification of IL-1β, IL-6, VEGF-A, MIP-1α, MIP-2, TNFα, KC, IL-17A and MCP-1 was carried out using an electrochemilumines-cence based immunoassay (MSD technology) according to the instructions provided in the kit. MSD plates were read using a Meso Sector S 600.

– Data handling and analysis.

The effect of the agents dosed via the i.t. route was evaluated against LPS-induced pulmonary inflammation (neutrophilia) at 4 and 24 h after the challenge. Data obtained are presented as mean +/- SEM and visualized in graphs. Statistical analysis was performed using GraphPad Prism 8 by One Way ANOVA followed by Dunnetts's test Multiple Comparison Test.

### Reporting summary

Further information on research design is available in the Nature Portfolio Reporting Summary linked to this article.

## Data availability

The data that support the findings of this study is available within the main text and its Supplementary Information file. Data is also available from the corresponding author upon request. The RNA sequencing of A549 epithelial cell data generated in this study have been deposited and are available in the NCBI Gene Expression Omnibus database under accession code GEO Submission GSE182853. The RNA sequencing of *Pseudomonas aeruginosa* data generated in this study have been deposited and are available in the NCBI Gene Expression Omnibus database under accession code GEO Submission GSE182847. The optimized structure data for all compounds generated in this study have been deposited as Gaussian input files and are available in Borealis database under accession code VF7T7J [https://doi.org/10.5683/SP3/VF7T7J].

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

## Acknowledgements

The authors would like to acknowledge funding from the Academy of Finland through its Centre of Excellence Programme Life-Inspired Hybrid Materials (LIBER grant 346105, 346109, 346110, 2022-2029: C.J., E.O., M.A.K., R.H.A.R., M.B.L.) and Academy Projects (grant 272578: and 272579: C.J., E.O., M.B.L., N.K.B., R.H.A.R.), as well as Research to Business funding from Business Finland (project numbers 6764/31/2019: C.J., E.O., M.B.L. and 6768/31/2019: K.L., P.S.). We acknowledge the support from Oakland University, MI, USA. We also want to acknowledge the Natural Sciences and Engineering Research Council of Canada (grant 2018-06338: S.M.T., J.F.T.), the Canadian Tricouncil (NFRFE-2018-00075: S.M.T., J.F.T). S.M.T. and J.F.T. wish to recognize that this work was made possible by the facilities of the Shared Hierarchical Academic Research Computing Network (SHARCNET: www.sharcnet.ca) and Compute/Calcul Canada. RNAseq sequencing service was provided by the Biomedicum Functional Genomics Unit at the Helsinki Institute of Life Science and Biocenter Finland at the University of Helsinki. We want to thank Dr. A. Bragonzi for the CF clinical isolates of *P. aeruginosa*. We also would like to thank Dr. A. Pätäri-Sampo for the clinical isolates of *P. aeruginosa* and *A. baumannii*, from the Helsinki University Hospital (HUS). We like to thank S. Soidinsalo for the detailed discussions and input.

## Author contributions

Conceptualization, C.J., N.K.B., E.O., R.H.A.R. and M.B.L.; Methodology, C.J., E.A.P., N.K.B., E.O, K.L., D. F., S.M.T. and J.F.T.; Analysis, C.J., N.K.B., E.O., K.L., S.M.T., E.A.P., M.A.K., D.F., J.F.T., R.H.A.R., P.S. and M.B.L; Writing, C.J., N.K.B., J.F.T., R.H.A.R., K.L. and M.B.L.; Review and editing, J.F.T., R.H.A.R., P.S. and M.B.L.

## Competing interests

Aalto University together with the University of Helsinki has filed for patent applications FI20185841A1, FI20205369A1 and FI20205368A1 on the effects of macrocycles on Gram-negative pathogens, with C.J., N.K.B., E.O., K.L., R.H.A.R., P.S. and M.B.L as co-inventors. C.J., E.O. and K.L. are co-founders of Arivin therapeutics Oy. The funders had no influence on the design or collection of results, interpretation of the data or writing of the manuscript. The remaining authors declare no other competing interests.
