## [Peer Review File · Nature Communications]

Repurposing host-guest chemistry to sequester virulence and eradicate biofilms in multidrug resistant *Pseudomonas aeruginosa* and *Acinetobacter baumannii*Reviewers' comments:

Reviewer #1 (Remarks to the Author):

The authors reported the interactions of a water-soluble Pillar[5]arene with homoserine lactones (HSLs) and lipopolysaccharides (LPS). Although authors have carried out lots of study presented in this manuscript, the key points are not sufficient to draw the conclusions. They concluded that the interaction between the water-soluble Pillar[5]arene and HSLs could shut down the virulence-controlled responses in bacterial pathogens. The interactions between the water-soluble Pillar[5]arene and LPS could affect the function of LPS in the outer membrane, resulting in increased penetration of antibiotics. As the authors claimed, the innovative nature of the presented work lies in the study of host-guest interactions of the water-soluble Pillar[5]arene with HSLs and LPS. However, after reading the discussion of the manuscript in detail, I came to the conclusion that the experimental details and their interpretation do not meet the standards required for publication. The data in fact does not support the main claim of the manuscript and I cannot recommend publication. At this stage this manuscript is of very poor quality and not publishable.

1) As the authors mentioned, a set of macrocycles, such as crown ethers, calixarenes, cucurbiturils, cyclodextrins and a water-soluble pillararene were used to clear the host-guest interactions of HSLs. I cannot see the logic of such a selection. There's nothing in common with these macrocycles. As I know, some of them can't be dissolved in water at 2.5 mM. Also the resources of them should be mentioned, at least in the SI.

2) The chemical structure of the "Pillar[5]arene" in Fig. 1 g) is incomplete, what the anions does it has. Bromide? Or chloride? And as I know this water-soluble trimethylamine functionalized pillar[5]arene is not commercially available. If it is from a co-worker, the basic characterizations should be listed in the SI, like ^1H NMRs, ^{13}C NMRs and Mass spectra.

3) In Fig. 2, part e, the stacked spectra of the water-soluble pillar[5]arene and HSLs, there are a set of peaks around 1.4 ppm, what is that? Some impurity or other solvent? As I know, there should be no peak of the trimethylamine functionalized pillar[5]arene's ^1H NMR around 1.4 ppm. Also the full stacked NMR spectra should be added in the SI, not just below 2 ppm.

4) P[5]a showed strong binding to the 3-oxo-C12 (64%) and 3:OH:C14 HSL (96%), both with prolonged acyl chains (Fig. 1b and e). However, there is different functional groups in the prolonged acyl chain between 3:OH:C14 from *Rhizobium leguminosarum* and 3-oxo-C12 from *Pseudomonas aeruginosa*. Do the functional groups influence on the host-guest interaction?

5) P[5]a significantly inhibited the synthesis of pyocyanin, and it was thought that P[5]a influenced on QS, which was verified by *E. coli* (Fig. S3E). Authors should further study the effect of P[5]a on the HSL changes of *Pseudomonas aeruginosa*.

6) LPS is an important component of bacterial outer membrane and has important physiological activity. Since P[5]a could significantly bind with LPS, thereby interfering with the structural arrangement of the bacterial outer membrane, and assisting the drugs enter into cells (Fig. 3k). why doesn't P[5]a affect other bacterial traits, such as growth traits (Fig. S2E)? Moreover, authors concluded no observable membrane-lytic activity of P[5]a treated cell A549, why? The membrane is composed of a variety of lipids and has a positively and negatively charged protein on the surface.

Reviewer #2 (Remarks to the Author):

The manuscript details the identification of a suitable host for Gram-negative bacterial quorum sensing molecules, acyl homoserine lactones (AHLs), through a fluorescence reporter strain based screen. Using six different categories of AHLs and hosts of varying cavity sizes, the authors have identified P[5]a as a suitable host, which shows strong interaction with the guest AHLs. With these results, the authors have proposed the use of this host as an antivirulent to tackle bacterial interactions. This system however does not effectively work against short acyl chain bearing AHLs. Thus, P[5]a will not be able to work against bacteria which produce such QSMs. Similarly, as virulent

bacteria often produce multiple classes of AHLs, inhibition of one of them would actually not reduce the virulence of the pathogen. While this macrocyclic host has not effect on bacterial growth, it does lead to the inhibition of virulence factor pyocyanin, as well as biofilm formation. The binding constant for different AHLs is calculated through dye displacement assay, and the binding is studied further through NMR and DFT calculations. Further, authors have extensively studied the antivirulence properties of P[5]a. The manuscript makes for an interesting read. However, there are some questions which need to be clarified by the authors before it can be considered for publication.

1. The flow cytometry studies with different reporter strains show that P[5]a selectively binds the 3-OH-C14:1 HSL. In that case, how can it act as a good antivirulent for other bacteria?
2. Similarly, one pathogen may produce multiple AHLs, with varying chain length. In that case, will P[5]a work as an antivirulent?
3. The energetics of host-guest interactions should be studied through techniques like isothermal titration calorimetry.
4. Authors have mentioned that P[5]a also proved effective against Δ LasR mutants (+), which have lost the ability to sense the 3-oxo-C12 HSL. If AHL is not sensed, then virulence must not be generated. How can this result be explained then?
5. The authors have observed that in one of the *P. aeruginosa* strains, there is no reduction in virulence? Can the authors comment on the probable reason for the same?
6. Authors are advised to provide checkerboard assay results for the combination of host and antibiotic.
7. Authors can perform the annexin assay to monitor reduction of apoptosis in mammalian cells.
8. Authors should study resistance development for the combination of P[5]a and antibiotic.
9. The cytotoxicity of the host should be calculated in terms of EC50 values.
10. Authors have tested activity of the antivirulent host in a mammalian cell line infection model. The protocol for the same should be provided.
11. Authors should perform in-vivo study validating this concept and antivirulent activity of P[5]a in a biofilm infection model.

Reviewer #3 (Remarks to the Author):

In the present work, the researchers have repurposed an existing macromolecular transport system for the sequestration of N-acyl-L-homoserine lactones, the most common QS system known to date. A chemistry grounded approach to limiting QS is not in itself novel as significant work in this area has been going on for over a decade nor do the authors make such a claim. The authors review of this area is rather limited and fails to cite crucial work done by Blackwell et al over the last two decades which were some of the first experiments attempting to modulate QS delivery.

However, the dual use nature of the P5a-type compounds to sequester AHLs and potentially limit other virulence factors simultaneously in a way which may reduce antibiotic resistance is important for the further development of QS-modulation as an effective combination strategy with existing antimicrobial approaches. Here the authors fail to note existing 'host-guest' type systems which have been employed to limit QS in the context of a Eukaryotic host by quorum quenching. Zeigler et al. should be added as a more recent reference for CDs that do this (see: *Chembiochem* . 2021 Apr 6;22(7):1292-1301. doi: 10.1002/cbic.202000773)

Overall, the authors have done an excellent job of developing this study from chemical theory to application. The development of the multiple sensor lines is of particular interest since it provides a tool for real-time visualization of the presence of multiple AHL types. This could be an important tool beyond this study alongside the modulating compounds themselves.

The development of their thought process is well-defined and makes the work relatively easy to follow and their claims are generally well-supported by the data provided. I find only minor issues:

1. The aforementioned references should be included for completeness.
- 2 Line 47: 'poses' seems awkward here
3. Line 99: I think the term 'downstream' might be more appropriate.
4. I'd recommend reviewing the document again for small wording discrepancies like #2 and #3 that appear throughout.

Reviewer 1

The authors reported the interactions of a water-soluble Pillar[5]arene with homoserine lactones (HSLs) and lipopolysaccharides (LPS). Although authors have carried out lots of study presented in this manuscript, the key points are not sufficient to draw the conclusions. They concluded that the interaction between the water-soluble Pillar[5]arene and HSLs could shut down the virulence-controlled responses in bacterial pathogens. The interactions between the water-soluble Pillar[5]arene and LPS could affect the function of LPS in the outer membrane, resulting in increased penetration of antibiotics. As the authors claimed, the innovative nature of the presented work lies in the study of host-guest interactions of the water-soluble Pillar[5]arene with HSLs and LPS. However, after reading the discussion of the manuscript in detail, I came to the conclusion that the experimental details and their interpretation do not meet the standards required for publication. The data in fact does not support the main claim of the manuscript and I cannot recommend publication. At this stage this manuscript is of very poor quality and not publishable.

Question 1. As the authors mentioned, a set of macrocycles, such as crown ethers, calixarenes, cucurbiturils, cyclodextrins and a water-soluble pillararene were used to clear the host-guest interactions of HSLs. I cannot see the logic of such a selection. There's nothing in common with these macrocycles. As I know, some of them can't be dissolved in water at 2.5 mM. Also the resources of them should be mentioned, at least in the SI.

Answer: We thank the reviewer for this comment. We understand that the manuscript was written with only little background on the logic of the selection of macrocycles. One reason for the very brief description of this was that the full characterization and analysis of properties left very little space within the word-limit set by the publisher. We understand that our brief description left many questions unanswered. The logic of selection is simply that the macrocycles represent a set of different types of molecular architectures, that we thought would show effects on HSL binding in our *E.coli*-based screening system. We aimed to study a set of widely diverging macrocycles with little in common. Rationally, we proposed that cavity size plays a defining role in the strength of the interactions. Previous publications on CDs suggested indeed that such interactions can occur. Although, since very little was known in general about HSL-macrocycle Host-Guest interactions, we wanted to study **1)** a very diverse set of macrocycles from differing classes, and **2)** a diverse set of HSLs. As with the diversity in macrocycles, the HSLs also included some of the shortest and longer HSL structures (C4-C14), structures containing carboxyl, hydroxyl and no functional groups, and one HSL that contained a phenol group.

We have now added text to clarify the rational, and logic behind the selection of the screen, again taking into account the limited space available.

- Page 3, line 75-76.
- Page 4, line 92-102.

We also understand that we had not in sufficient detail described the sources of the macrocycles, this description is now updated in the Supplementary Information and in

Material & Methods. Also added there are those macrocycles for which solubility was a problem, and DMSO had to be used to solubilize the macrocycles. Final concentrations of DMSO in *E. coli* reporter screening was always kept at 1%, to avoid impact on growth.

We have further addressed the reviewers' comments and clarified the manuscript:

- Supplementary table 1, last column.
- Materials & Methods, "Materials".
- Materials & Methods, "Fluorescent *E. coli* reporter system".

Question 2. The chemical structure of the "Pillar[5]arene" in Fig. 1 g) is incomplete, what the anions does it has. Bromide? Or chloride? And as I know this water-soluble trimethylamine functionalized pillar[5]arene is not commercially available. If it is from a co-worker, the basic characterizations should be listed in the SI, like 1H NMRs, 13C NMRs and Mass spectra.

Answer: We understand that this information was not clearly enough presented. The anion is bromide. The chemical structure of **P[5]a** has now been correctly updated. The **P[5]a** macrocycle was synthesized according to the reported procedure (Montes-Garcia, et al., Chem. Eur. J. 2014, 20, 8404 – 8409). We have added the synthesis scheme of **P[5]a** in the Supplementary Information (Supplementary Fig. 13). We have also included the 1H-NMR, 13C-NMR, COSY and HSQC spectra in the Supplementary Information (Supplementary Fig. 14,15,16).

We thank the reviewer for pointing this out and have further addressed the reviewers' comments and clarified the manuscript with the following additions:

- Page 5, Figure 1g.
- Supplementary Fig. 13, page 14.
- Supplementary Fig. 14a,b, page 15.
- Supplementary Fig. 15a,b, page 16.
- Supplementary Fig. 16a,b,c,d, page 17.

Question 3. In Fig. 2, part e, the stacked spectra of the water-soluble pillar[5]arene and HSLs, there are a set of peaks around 1.4 ppm, what is that? Some impurity or other solvent? As I know, there should be no peak of the trimethylamine functionalized pillar[5]arene's 1H NMR around 1.4 ppm. Also the full stacked NMR spectra should be added in the SI, not just below 2 ppm.

Answer: Our analysis shows that the purity of **P[5]a** is very high (Supplementary Fig. 14,15). The peak at 1.4 ppm is a volatile solvent and a very minor fraction (less than 1 mole), which is not present in the compound further dried overnight (Supplementary Fig. 16). We have further addressed the reviewers' comments and added the fully stacked NMR spectra to the Supplementary information:

- Supplementary Fig. 14a,b, page 15.
- Supplementary Fig. 15a,b, page 16.
- Supplementary Fig. 16a,b,c,d, page 17.

Question 4. P[5]a showed strong binding to the 3-oxo-C12 (64%) and 3:OH:C14 HSL (96%), both with prolonged acyl chains (Fig. 1b and e). However, there is different functional groups in the prolonged acyl chain between 3:OH:C14 from *Rhizobium leguminosarum* and 3-oxo-C12 from *Pseudomonas aeruginosa*. Do the functional groups influence on the host-guest interaction?

Answer: This is a very interesting question. We selected only naturally occurring HSL-molecules for our study. These vary not only with length of the acyl chain but also in some functional groups. The set of naturally occurring HSLs therefore represents a diverse group. We included the very short C4 HSL, and up to the much longer C14 HSL. One of the naturally occurring HSL contains even a phenol group (pC). In general, we observed a general trend in the binding affinity, which matches the length of the acyl chain, where longer acyl chains resulted in a stronger binding affinity (3OH-C14:1≥3-oxo-C12>3-oxo-C8>>3-oxo-C6~pC~C4). The order of chain length is striking. Modeling further suggested that the long hydrophobic acyl chains reside in P[5]a's internal cavity, making this the most stable complex. The presence of either carboxyl, hydroxyl or no functional groups did not appear to have a major influence on the binding fit, apart from affecting the total length of the hydrophobic acyl chains, but since we logically restricted to relevant naturally occurring HSLs it was not experimentally studied how such variations would have affected. A systematic study of functional groups would indeed be interesting, but the restriction to naturally occurring ones is in our minds logical, which we hope the reviewer agrees with.

We have further addressed the reviewers' comments and clarified the manuscript:

- Page 10-11, Line 264-268.

Question 5. P[5]a significantly inhibited the synthesis of pyocyanin, and it was thought that P[5]a influenced on QS, which was verified by *E. coli* (Fig. S3E). Authors should further study the effect of P[5]a on the HSL changes of *Pseudomonas aeruginosa*.

Answer: We agree that characterization of *P. aeruginosa* is essential, and the interaction between P[5]a and the *P. aeruginosa* QS C4 and 3-oxo-C12 HSL were studied thoroughly. In addition to the *E. coli* verification, these interactions were then further characterized, and binding affinities determined through dye displacement (Fig. 2b,c,d). These results were in line with the *E. coli* reporter results. This was then followed with NMR (Fig. 2e) and high-resolution modeling (Fig. 2f,g). We then turned to the effects on *P. aeruginosa* (PAO1). We observed dose-dependent downregulation of known QS responses, the toxin pyocyanin and biofilm production (Fig. 1k,l respectively). We also demonstrated that the effects on pyocyanin production in *P. aeruginosa* very closely followed the concentration dependence

to the fluorescence signal in the *E. coli* biosensor (Supplementary Information Fig. 3e). Importantly, we performed mRNA sequencing on **P[5]a** treated *P. aeruginosa*. Gene set enrichment showed that the addition of **P[5]a** resulted in a downregulation of QS responses. Finally in Fig. 5a, we challenged A549 epithelial cells with purified *P. aeruginosa* 3-oxo-C12 HSL. The 3-oxo-C12 can diffuse freely through bacterial membranes is used as signalling tools, however in mammalian cells, the 3-oxo-C12 interacts with bilayer membranes, disrupting the lipid domains. This results in the release of pro-inflammatory cytokines, TNFR1 signalling lymphocyte cell death, and an exacerbation of airway inflammation. We demonstrate that the interaction between **P[5]a** and 3-oxo-C12 sequesters the 3-oxo-C12 HSL, protecting A549 epithelial cells against cell death responses by 3-oxo-C12. The data in these experiments all supported strongly the conclusion that the effect of **P[5]a** *P. aeruginosa* is mediated through HSL interactions.

Question 6. LPS is an important component of bacterial outer membrane and has important physiological activity. Since P[5]a could significantly bind with LPS, thereby interfering with the structural arrangement of the bacterial outer membrane, and assisting the drugs enter into cells (Fig. 3k). why doesn't P[5]a affect other bacterial traits, such as growth traits (Fig. S2E)? Moreover, authors concluded no observable membrane-lytic activity of P[5]a treated cell A549, why? The membrane is composed of a variety of lipids and has a positively and negatively charged protein on the surface.

Answer: We thank the reviewer for raising this interesting question, which we have extensively studied throughout the manuscript. The outer membrane of Gram-negative bacteria comprises an asymmetric bilayer, with glycerophospholipids in the inner leaflet and lipopolysaccharide (LPS) in the outer leaflet. This unique permeability barrier protects the bacteria from toxic environmental factors (such as antibiotics). The inner leaflet of the bacterial OM shares similarity with eukaryotic bilayer membranes, with glycerophospholipids present in both membranes.

First, we ruled out whether **P[5]a** effects the outside of the bacterial OM, or also the cytoplasmic membranes, through several viability assessments. Firstly, Figure 1f shows that **P[5]a** does not affect the bacterial cytoplasmic membranes. No changes in growth rate, or bacteriostatic/bactericidal effects were observed in the concentration ranges up to 2.5 mM **P[5]a**. Next, we aimed to investigate whether **P[5]a** interacts with the bacterial OM outer leaflet, where LPS is inserted, or with the OM inner leaflet, consisting of glycerophospholipids. To study this, we further investigated the interactions between **P[5]a** and purified LPS (Fig. 3a,b,c,d). Next, we turned to A549 epithelial cells, of which the eukaryotic bilayer membranes also consists of glycerophospholipids (similar as the bacterial OM inner leaflet), using a trypan blue staining. **P[5]a** displayed low toxicity towards A549 cells, with no observable membrane-lytic activity (Fig. 5a) We further studied the effects of **P[5]a** eukaryotic membranes through a cytotoxicity assessment and mRNA sequencing of A549 epithelial cells treated with varying concentrations of **P[5]a** (Fig 5c,d,e). This further suggests that the interactions with the

bacterial OM occur through interactions in the membrane outer leaflet, where LPS is inserted, rather than impacting the membrane inner leaflet consisting of glycerophospholipids.

Other studies have also described LPS targeting treatments that target LPS and disrupt the Outer Membrane, but lack the ability to disrupt cytoplasmic bacterial and eukaryotic membranes (hence without lytic effects):

- SPR741, is a polymyxin analogue that functions as an antibiotic potentiator that retains the ability to disrupt the bacterial OM, but lacks the capability to depolarize the cytoplasmic membranes.
- Pentamidine is an antibiotic sensitizer in Gram-negative pathogens that interacts with LPS. Pentamidine lacks a bacterial MIC, and is also <https://www.nature.com/articles/nmicrobiol201728>

There is also an example of an LPS targeting treatment that target LPS and disrupts the Outer Membrane and BamA, the β -barrel folding complex (BAM) that is required for the folding and insertion of β -barrel proteins into the outer membrane of Gram-negative bacteria, but lacks the ability to affect eukaryotic membranes (without lytic effects):

- The novel outer membrane protein targeting antibiotics (OMPTAs) are bactericidal and have a mechanism of action that involves binding to both lipopolysaccharide and the main component (BamA) of the β -barrel folding complex (BAM) that is required for the folding and insertion of β -barrel proteins into the outer membrane of Gram-negative bacteria (Extended data Fig. 3) <https://www.nature.com/articles/s41586-019-1665-6.pdf>

To conclude, the data in these experiments all supported the hypothesis that **P[5]a** interacts with LPS in the bacterial OM outer leaflet. Furthermore, these results are in line with other treatments in which interactions with LPS are either a known or suggested part of the mechanism of action.

We have further addressed the reviewers' comments and clarified the manuscript:

- Page 16, line 416-426.
- Page 17, line 449-453.
- Page 21, line 565-576.

Reviewer 2

The manuscript details the identification of a suitable host for Gram-negative bacterial quorum sensing molecules, acyl homoserine lactones (AHLs), through a fluorescence reporter strain based screen. Using six different categories of AHLs and hosts of varying cavity sizes, the authors have identified P[5]a as a suitable host, which shows strong interaction with the guest AHLs. With these results, the authors have proposed the use of this host as an antivirulent to tackle bacterial interactions. This system however does not effectively work against short acyl chain bearing AHLs. Thus, P[5]a will not be able to work

against bacteria which produce such QSMs. Similarly, as virulent bacteria often produce multiple classes of AHLs, inhibition of one of them would actually not reduce the virulence of the pathogen. While this macrocyclic host has not effect on bacterial growth, it does lead to the inhibition of virulence factor pyocyanin, as well as biofilm formation. The binding constant for different AHLs is calculated through dye displacement assay, and the binding is studied further through NMR and DFT calculations. Further, authors have extensively studied the antivirulence properties of P[5]a. The manuscript makes for an interesting read. However, there are some questions which need to be clarified by the authors before it can be considered for publication.

Question 1. The flow cytometry studies with different reporter strains show that P[5]a selectively binds the 3-OH-C14:1 HSL. In that case, how can it act as a good antivirulent for other bacteria?

Answer: The effects of P[5]a on a microbial community are described in figure 1 h,i,j. These experiments further demonstrate that P[5]a preferentially binds to HSLs with a prolonged carbon chain. And even in a more complex environment with a variety of microorganisms, this preference is still observed. The selectivity, we propose, can often be a beneficial trait, to target pathogens more specifically.

For example, a major problem with current oral antibiotics, is the large impact of the antibiotic on the human microbiome. Gut microbiota play an important role in health of the host. When antibiotics are taken orally, part of the dose is not absorbed and reaches the gastro-intestinal tract, where the antibiotic can have a detrimental impact on the microbiome diversity, affecting the overall health of the host. This decimation of the microbiome flora in turn increases chances for *Clostridium difficile* infections, which mostly occur during or shortly after antibiotic regimens. The effects of antibiotics on gut bacteria is further described in an excellent paper in Nature: <https://www.nature.com/articles/s41586-021-03986-2>

Hence, this provides strong motivation for novel treatments with a more targeted “narrow-spectrum” approach, even in complex microbial communities. As such, P[5]a would see use as a narrow-spectrum treatment.

We have now added text to clarify the rational:

Page 7, line 171-175.

Question 2. Similarly, one pathogen may produce multiple AHLs, with varying chain length. In that case, will P[5]a work as an antivirulent?

Answer: Indeed, for instance the pathogen *P. aeruginosa* utilizes two different HSLs, the prolonged 3-oxo-C12 HSL and the shorter C4 HSL. In *P. aeruginosa*, the QS systems are organized in a hierarchy, with the LasR receptor (which detects the 3-oxo-C12 HSL), at the top of the hierarchy. The 3-oxo-C12-LasR complex leads to auto-induction 3-oxo-C12 synthesis, as well as expression of the C4 HSL and cognate RhIR receptor. Given the importance of 3-

oxo-C12 – LasR at the top of the hierarchy, strategies that target the 3-oxo-C12 HSL could effectively “shut down” the whole quorum-sensing response cascade.

We observed the effects of P[5]a on a widespread set of *P. aeruginosa* isolates, and found the treatment significantly effective in most cases (>93%) (Fig. 1m, Fig. 4a). This indicates that the 3-oxo-C12 is an important and effective target for anti-virulence strategies against *P. aeruginosa*.

Another example of a pathogen that also utilizes several different HSLs, is the Top Priority *Acinetobacter baumannii*. *Acinetobacter baumannii* utilizes multiple prolonged carbon-chain HSLs (from a C10 to C16). We also observed good effectivity of P[5]a on the three clinical isolates (Fig. 1n).

So, if HSLs with a prolonged carbon chain play an important role in the quorum sensing of the bacterial pathogen, P[5]a could be an effective anti-virulence strategy.

We have now added text to clarify the rationale:

- Page 21, line 546-548.

Question 3. The energetics of host-guest interactions should be studied through techniques like isothermal titration calorimetry.

Answer: We have extensively explored Isothermal Titration Calorimetry (ITC) to determine the energetics of host-guest interactions. However, the poor solubility of 3-oxo-C12 and 3-OH-C14 HSLs in H₂O, required large quantities of DMSO solvent. For this reason, we also used a dye-displacement method to determine affinities. We found that the dye-displacement is a highly reliable method to accurately determine the affinities for all the host-guest interactions between P[5]a and the set of HSLs. To save space in the manuscript we chose to focus on the dye-displacement assay.

The ITC shows that the binding event is enthalpically driven and spontaneous at 310 K. Complex formation between any combination of HSL and receptor is spontaneous ($\Delta G < 0$) at the experimental temperature (310 K). All interactions with P[5]a were enthalpically driven. The most favorable interactions with P[5]a were observed for 3-OH-C14, 3-oxo-C12, and 3-oxo-C6 HSLs. Results from ITC reveals less affinity for the shorter HSLs. These results indicate a clear preference towards specific HSLs. The data are shown below in figures R1 and R2 (unpublished data and not part of our submitted manuscript) and verify the results obtained in the dye-displacement assays.

Figure R1. ITC experiments at 310 K in $\text{H}_2\text{O}/\text{DMSO}$ (30% v/v) of (a) LuxI@P[5]a, (c) Las@P[5]a and (e) Cin[P[5]a. ITC experiments at 310 K in LB/DMSO (30% v/v) of (b) LuxI@P[5]a, (d) Las@P[5]a and (f) Cin[P[5]a.

Figure R2. ITC experiments at 310 K in $\text{H}_2\text{O}/\text{DMSO}$ (30% v/v) of (a) Rhl@P[5]a, (c) Rpa@P[5]a. ITC experiments at 310 K in LB/DMSO (30% v/v) of (b) Rhl@P[5]a, (d) Rpa@P[5]a. The results show unspecific binding and some data could not be fitted.

Question 4. Authors have mentioned that P[5]a also proved effective against ΔLasR mutants (+), which have lost the ability to sense the 3-oxo-C12 HSL. If AHL is not sensed, then virulence must not be generated. How can this result be explained then?

Answer: The panel of *P. aeruginosa* clinical isolates (Fig. 4a) consisted of a wide range of clinical isolates, including 10 ΔLasR mutants. Although QS inhibitors block the production of 3-oxo-C12 HSL autoinducer and the formation of biofilm, deletion of the gene that encodes the LasR receptor does not prevent biofilm development. In these cases, a phosphate starvation protein PhoB is known to override the necessity for LasR through the activation of IQS (2-(2-hydroxyphenyl)-thiazole-4-carbaldehyde) production. In turn, IQS activates the expression of the pqs genes, which produces the additional required autoinducer through activation of rhl expression.

In this study, we did not include the interactions between **P[5]a** and the other two known QS systems in *P. aeruginosa*: 2-heptyl-3-hydroxy-4-quinolone (PQS) and IQS. Since both IQS (and to a lesser extent PQS) play an important role in QS responses in Δ LasR mutants, further studies would have to elucidate the possible interactions. However, these studies are beyond the scope of this study. A full overview of the QS systems, and the hierarchy, is described in the following Nature Reviews microbiology paper: <https://www.nature.com/articles/nrmicro.2016.89>

Question 5. The authors have observed that in one of the *P. aeruginosa* strains, there is no reduction in virulence? Can the authors comment on the probable reason for the same?

Answer: We did not observe either phenotypical characteristics or documented characteristics of the clinical isolate (attached excel) that could help clarify the lack of effectivity against the “rebel” strain. However, overall we observed widespread effectivity (>93% statistically significant) of **P[5]a** against extensively antibiotic resistant isolates, CF specific isolates, and general IATS serotypes.

Question 6. Authors are advised to provide chequerboard assay results for the combination of host and antibiotic.

Answer: We thank the reviewer for this comment and good suggestion. In several places we have explored combinatory and synergistic effects between **P[5]a** and a variety of antibiotics with differing mechanisms of action (Fig. 3, Fig. 4). We also further explored the proposed mechanism behind the observed synergy with antibiotics (Fig. 2, Fig. 3, Fig. 6). We agree that the chequerboard assay would be a good method to determine the interplay between the observed synergy and it would certainly be of interest for follow-up studies. While additional insight on antibiotics interactions certainly is of interest, we have focused here on establishing the fundamental framework for the action of **P[5]a**, starting from host-guest chemistry, to molecular biology, microbiology, medical biology and drug development. The wide range of approaches for this set of questions of course leave still room for in depth studies. Within the limitations of the overall size of the manuscript we have focused on critical experiments to establish the connection between host and antibiotic. We hope that the wide approach that we have taken is appreciated, and that filling in the gaps on understanding on antibiotics resistance for example will be the topic of follow-up studies.

Question 7. Authors can perform the annexin assay to monitor reduction of apoptosis in mammalian cells.

Answer: We studied the viability of A549 epithelial cells with a Trypan Blue viability dye exclusion test (now Figure 5a). We also performed a Celltox Green Cytotoxicity assay of A549 epithelial cells, untreated, treated with **P[5]a**, infected with PAO1, or both. mRNA sequencing was used to further characterize possible apoptosis responses in A549 epithelial cells, untreated, treated with **P[5]a**, infected with PAO1, or both.

We have addressed the A549 epithelial cell viability and cytotoxicity in:

- Fig. 5a,b,c.
- Page 16,17, line 414-126.

Question 8. Authors should study resistance development for the combination of P[5]a and antibiotic.

Answer: The resistance development for the combination of P[5]a and four different clinically relevant antibiotics with differing mechanisms of action, is described in Fig. 3f, g, h and i for the antibiotics Aztreonam, Cefepime, Meropenem and Tobramycin respectively. In this experiment, we start to culture *P. aeruginosa* (in 96 WP, liquid) with a range of different antibiotic concentrations, and then we measure what is the highest antibiotic concentration to survive. This culture then gets re-cultured with all the ranges of antibiotic, and we measure what is the highest concentration again. We do this for 14 days in a row. The antibiotic concentrations, as well as the definitions of susceptible (below yellow line), intermediate susceptible (between yellow and red) and resistant (above the red line), are defined by the “Clinical & Laboratory Standards Institute” (CLSI).

We have addressed the resistance development for the combination of P[5]a and antibiotic in:

- Fig. 3f,g,h,i,j.
- Page 14, line 338-357.

Question 9. The cytotoxicity of the host should be calculated in terms of EC50 values.

Answer: We performed cytotoxicity assessments of the effects of P[5]a on A549 cells up to highest concentration we included in the mechanism of action studies we performed in vitro; 2.5 mM P[5]a. First, we observed in a trypan blue cell viability assessment (now Figure 5. c zoom-in and Figure 5. d), that the highest tested concentration of 2.5 mM P[5]a did not result in observable differences in cell viability. Next, we performed mRNA sequencing of A549 cells exposed to 100 μ M and 2.5 mM P[5]a, which confirmed that P[5]a was well tolerated, resulting in only small deviations from the untreated control.

We have addressed the viability/cytotoxicity of P[5]a on A549 cells in:

- Fig. 5a,b,c,d,e,f.
- Line 414-466.

Question 10. Authors have tested activity of the antivirulent host in a mammalian cell line infection model. The protocol for the same should be provided.

Answer: The method was previously described under the header “Cytotoxicity assay”. However, after the good suggestion from the reviewer, we have now given the method its own chapter called “Mammalian cell line infection model”.

We have addressed the reviewers' comments in:

- Materials & Methods, "Cytotoxicity assay" and "Mammalian cell line infection model".

Question 11. Authors should perform in-vivo study validating this concept and antivirulent activity of P[5]a in a biofilm infection model.

Answer: We acknowledge the reviewer's suggestion to validate our anti-virulence strategy in an *in vivo* proof-of-concept study.

To address this suggestion, we validated the efficacy of **P[5]a** in a LPS-induced lung inflammation model in mice. In this model, C57BL/6J mice are challenged intranasally with LPS (purified from *P. aeruginosa*, 10 µg/kg), where the LPS-challenge induced an increase in white blood cells and neutrophils in the lung, as well as an increase in the inflammatory mediators IL-1β, IL-6, IL17A, KC, TNFα, MCP-1, MIP-1α and MIP-2 and the angiogenic factor VEGF-A.

The aim of this study was to investigate the effects of intratracheally (i.t.) administered **P[5]a** at three dose levels (0.1 mg/kg, 0.5 mg/kg & 3 mg/kg) on white blood cells, neutrophils and inflammatory mediators in the lung after the LPS challenge.

- **Aim:** Evaluate if **P[5]a** -LPS interaction occurs in vivo and whether the interaction lowers inflammatory responses, measured by inflammatory cytokine analysis and neutrophil/white blood cell recruitment.
- **Results:** **P[5]a** significantly reduced key inflammatory cytokines, KC (CXCL1), TNF-α, MIP-2 (CXCL2) and IL-6, as well as a 30 & 31% reduction in white blood cells and neutrophils respectively, after 24 hours at 3 mg/kg dose of P[5]a.

Our motivation for choosing this model was because it further demonstrates the effects of **P[5]a** as a direct anti-virulence treatment. This mechanism of action is wholly differentiated from standard-of-care antibiotics, which only target pathogens viability. These results also further confirm the anti-virulence effects of **P[5]a** in the mammalian cell line infection model (Fig. 6. a,b). Results are described in an additional chapter and figure 6.

We have addressed the reviewers' comments in:

- Fig. 6c,d,e,f,g,h,i.
- Page 20, line 502-524
- Page 22, line 603-607.

Reviewer 3

In the present work, the researchers have repurposed an existing macromolecular transport system for the sequestration of N-acyl-L-homoserine lactones, the most common QS system known to date. A chemistry grounded approach to limiting QS is not in itself a novel or significant work in this area has been going on for over a decade nor do the authors make

such a claim. The authors review of this area is rather limited and fails to cite crucial work done by Blackwell et al over the last two decades which were some of the first experiments attempting to modulate QS delivery.

However, the dual use nature of the P5a-type compounds to sequester AHLs and potentially limit other virulence factors simultaneously in a way which may reduce antibiotic resistance is important for the further development of QS-modulation as an effective combination strategy with existing antimicrobial approaches. Here to the authors fail to note existing 'host-guest' type systems which have been employed to limit QS in the context of a Eukaryotic host by quorum quenching. Zeigler et al. should be added as a more recent reference for CDs that do this (see: *Chembiochem* . 2021 Apr 6;22(7):1292-1301. doi: 10.1002/cbic.202000773)

Overall, the authors have done an excellent job of developing this study from chemical theory to application. The development of the multiple sensor lines is of particular interest since it provides a tool for real-time visualization of the presence of multiple AHL types. This could be an important tool beyond this study alongside the modulating compounds themselves.

The development of their thought process is well-defined and makes the work relatively easy to follow and their claims are generally well-supported by the data provided. I find only minor issues.

Answer: We have now updated the reference list to better represent important previous work related to the sequestration of quorum-sensing. We have also expanded this topic more in the introduction to better reflect those previous endeavors.

Question 1. The aforementioned references should be included for completeness.

Answer: Thank you for the good suggestions, the references have been included. We have now also expanded on the previous efforts and publications, as motivation for the E. coli based HSL reporter system.

We have addressed the reviewers' comments in:

- Page 3, line 56-58.
- Page 3, line 71-76.
- Page 4, line 93-99.

The following references have been included also to address the reviewers' comments and provide a more complete overview of previous efforts:

- References 8, 9, 24, 25.

Question 2. Line 47: 'poses' seems awkward here.

Answer: Adapted.

Question 3. Line 99: I think the term 'downstream' might be more appropriate.

Answer: Changes implemented.

Question 4. I'd recommend reviewing the document again for small wording discrepancies like #2 and #3 that appear throughout.

Answer: We thank the reviewer for this comment. We have now made small adaptations in the manuscript, considering the above-mentioned good suggestions, to improve the readability.

REVIEWERS' COMMENTS

Reviewer #1 (Remarks to the Author):

The revision of authors and the clarifications in the rebuttal letter fully addressed this reviewer's concern. I would like recommend the publication of the manuscript in Nature Communications.

Reviewer #2 (Remarks to the Author):

The authors have very clearly and satisfactorily addressed all the queries, and have additionally performed various experiments wherever required. The revised manuscript is recommended for publication, with just the following minor suggestions for authors.

1. In the reply to the first query, authors have spoken about the selectivity of antimicrobial therapy in the context of gut microbiome. Can the authors comment on the systems of quorum sensing observed in gut microbiota and if they will be non-susceptible to this therapy?
2. The ITC experiments show the affinity of the host for longer chain HSLs. The authors can consider including this data as a part of the supporting information.

Reviewer #1 (Remarks to the Author):

The revision of authors and the clarifications in the rebuttal letter fully addressed this reviewer's concern. I would like recommend the publication of the manuscript in Nature Communications.

Reviewer #2 (Remarks to the Author):

The authors have very clearly and satisfactorily addressed all the queries, and have additionally performed various experiments wherever required. The revised manuscript is recommended for publication, with just the following minor suggestions for authors.

1. In the reply to the first query, authors have spoken about the selectivity of antimicrobial therapy in the context of gut microbiome. Can the authors comment on the systems of quorum sensing observed in gut microbiota and if they will be non-susceptible to this therapy?

Several studies have investigated the role of quorum sensing utilized by microorganisms present in the gut ecosystem. The characterized HSLs that bacteria utilize in this environment (for instance the 3-oxo-C12:2-HSL) are highly identical to the HSLs we explored in this study and with which we observed large affinity. However, follow-up studies are required to further investigate the ability of P[5]a to bind HSLs within the gut microbiome environment and the potential of this interaction to reduce inflammation, as is common in for instance inflammatory bowel syndrome.

Coquant G, Aguanno D, Brot L, Belloir C, Delugeard J, Roger N, Pham HP, Briand L, Moreau M, de Sordi L, Carrière V, Grill JP, Thenet S, Seksik P. 3-oxo-C12:2-HSL, quorum sensing molecule from human intestinal microbiota, inhibits pro-inflammatory pathways in immune cells via bitter taste receptors. *Sci Rep.* 2022 Jun 8;12(1):9440. doi: 10.1038/s41598-022-13451-3. PMID: 35676403; PMCID: PMC9177545.

2. The ITC experiments show the affinity of the host for longer chain HSLs. The authors can consider including this data as a part of the supporting information.

As we previously commented, we found in the dye-displacement a more reliable method to accurately determine the affinities for all the host-guest interactions between P[5]a and the set of HSLs.

The binding event is enthalpically driven and spontaneous at 310 K. Complex formation between any combination of HSL and receptor is spontaneous ($\Delta G < 0$) at the experimental temperature (310 K). All interactions with P[5]a were enthalpically driven. The most favorable interactions with P[5]a were observed for 3-OH-C14, 3-oxo-C12, and 3-oxo-C6 HSLs. Results from ITC reveals less affinity for the shorter HSLs. These results indicate a clear preference towards specific HSLs. However, limited solubility forced us to use a large amount of DMSO to get reliable data.

We concluded that ITC is not ideal to investigate the host-guest affinity because of the distortions that arise from solubility problems. These are inherent to the system and could not be resolved. On the other hand, the dye displacement was reliable. Since the ITC data does not add more understanding to the interactions, and to avoid confusion for readers, we felt it troublesome to add the ITC results. We hope the reviewer understands our concerns.